# Critical node identification and resilience analysis against cascading failures

Anqi Liu[1]*, Wenfu Zhao[2]

**1** School of Energy and Transportation Engineering, Inner Mongolia Agricultural University, Hohhot, Inner Mongolia, China, **2** School of Civil and Transportation Engineering, Shenzhen University, Shenzhen, Guangdong, China

\* 1025870083@qq.com

## Abstract

Ensuring the robustness and resilience of critical infrastructure networks such as transportation and energy systems is a core security challenge for modern societies. Vulnerabilities in these networks often concentrate on a small number of critical nodes, whose failure can trigger catastrophic cascading failures. Therefore, accurately identifying critical nodes and formulating effective reinforcement strategies are crucial for enhancing the overall defense capability of the system. Existing graph neural network (GNN)-based methods often rely on topological centrality metrics, neglecting the distribution of node information and the impacts of cascading failures. To bridge this gap, this study constructs a comprehensive analytical framework (TEC-GNN, Topology-Entropy-Cascading Graph Neural Network) integrating graph neural networks, feature engineering, and resilience assessment. It aims to address two core questions: which graph neural network model is most suitable for critical node identification, and how to enhance network resilience by regulating redundant resource allocation. Systematic evaluation indicates that the GraphSAGE model delivers the best overall performance in critical node identification. Its results exhibit high consistency with supervised signals (Spearman's correlation coefficient of 0.822), achieving a Normalized Discounted Cumulative Gain at Top-K (NDCG@K) of 0.918, an F1 Score at Top-K (F1@K) of 0.879, and a Top-K accuracy of 0.879. Its inference efficiency (0.002 s) is comparable to GCN and significantly outperforms GAT, meeting the demands of real-time analysis for large-scale networks. After feature dimension reduction via principal component analysis (PCA), the model's discriminative power further improved, with effect size (Cohen's d) increasing by approximately 4% without efficiency loss, validating the effectiveness of scientific dimension reduction. The model's accuracy was robustly validated through attack experiments: selectively removing the top 10% critical nodes identified by GraphSAGE reduced the network's largest connected component ratio (LCC_Ratio) to approximately 0.4, severely impairing network functionality. When the removal rate reached 20% (equivalent to 60% removal in random attacks), the network became nearly paralyzed. Another core finding reveals the complex nonlinear

**Data availability statement:** The complete reproducibility package, including all source code, hyperparameter configurations, preprocessing scripts, and environment specifications, is currently held in a private repository to avoid premature disclosure during the peer review process. Upon acceptance of this manuscript, the repository will be made publicly available at: https://gitee.com/anqi_Liu666/paper-reproduction-2025. A detailed README document outlining the structure and usage is provided in the Supplementary Materials.

**Funding:** This research was supported by the Natural Science Foundation of Inner Mongolia Autonomous Region (grant numbers 2025QN05124 and 2024QN07001) to Anqi Liu. The funders had no role in study design, data collection and analysis, decision to publish, or preparation of the manuscript.

**Competing interests:** The authors have declared that no competing interests exist.

regulatory mechanism of redundancy coefficient $\beta$ on network resilience. The resilience metric $R$ exhibits clear diminishing marginal returns with increasing $\beta$: $R$ rises rapidly as $\beta$ increases from 0 to 0.5, then slows significantly with fluctuations thereafter. Based on this, the study proposes a "precision reinforcement" strategy: enhancing redundancy allocation only for critical nodes identified by GraphSAGE enables low-cost resilience improvement (e.g., $R$ increases from 0.874 to 0.883). This strategy provides an efficient path for system fortification under resource constraints. The research framework proposed in this paper provides interpretable and scalable theoretical and methodological support for vulnerability assessment and resilience enhancement in critical infrastructure. The validated GraphSAGE model and "targeted reinforcement" strategy are particularly suitable for risk prevention and resource optimization in major infrastructure systems requiring dynamic analysis and rapid response, such as transportation and power grids.

## Introduction

The structural complexity and high interdependency of infrastructure networks render them vulnerable to cascading failures triggered by localized disruptions. Owing to the dense interconnectivity among critical infrastructure systems—including power grids, transportation networks, and communication systems—localized failures can propagate into large-scale system paralysis through cascading effects, as demonstrated by the 2003 Northeast Blackout and 2021 Texas Power Grid Collapse [1,2]. Initial fault propagation via topological pathways progressively degrades system performance, ultimately resulting in cross-regional breakdowns (e.g., grid outages, supply chain disruptions) [3,4]. Analysis of cascading failure mechanisms enables precise identification of high-risk infrastructure components, informing protective strategies such as redundancy optimization and dynamic regulation. This approach significantly enhances network resilience, with critical implications for safeguarding socio-economic stability.

The functional reliability of infrastructure networks (e.g., transportation, power systems) hinges significantly on critical-node facilities [5,6]. Compared with edge failures, node failures present a higher risk of extensive cascading failures. Nodes serve as crucial hubs (e.g., transportation hubs, substations) tasked with resource processing and transfer. Any disruption to these nodes undermines connectivity and exacerbates fault propagation. Empirical instances, such as the 2021 Suez Canal obstruction that disrupted regional supply chains and multiple large-scale blackouts in California frequently triggered by critical node failures, underscore the elevated risk associated with node vulnerabilities [2,7]. Resilience evaluations indicate that protective strategies focused on critical nodes (e.g., redundancy configuration, dynamic routing optimization) typically yield greater cost-effectiveness in enhancing network robustness than edge-based methods [8]. Therefore, precise identification of critical nodes is indispensable for formulating effective risk mitigation policies and designing robust systems.

As a core challenge in network science, the identification of critical nodes aims to pinpoint those nodes that are crucial for network stability and functionality. With its applications spanning social networks, bioinformatics, and transportation systems, this field has been predominantly advanced through two paradigms: traditional topology – based methods that rely on graph centrality metrics [9,10], and contemporary data-driven approaches that utilize machine learning, specifically graph neural networks (GNNs) [11,12]. Although both paradigms offer certain advantages, they also present specific limitations, such as restricted scalability, poor interpretability, or insufficient integration of multi – dimensional node and structural features. This paper addresses this gap by proposing a TEC-GNN (Topology-Entropy-Cascading Graph Neural Network) architecture, which integrates structural topology with multi-source node attributes (including information entropy and cascade impact metrics) to enable comprehensive characterization of node influence. This framework enhances network resilience and informs optimal strategies to mitigate cascading failures.

The resilience of complex networks, which underpin critical infrastructure and technological systems, has become essential for ensuring their functionality and survival in the face of disruptions. However, existing assessment methods, which rely on the accurate identification of key nodes, predominantly employ single metrics such as robustness [13], node protection [14], or cascade suppression [15]. These unidimensional approaches offer limited perspectives and fail to fully capture the multifaceted nature of resilience. To address these limitations and establish a systematic, theoretically grounded assessment framework, this study adopts the taxonomic perspective on network resilience proposed by Artime et al. [16]. This framework deconstructs resilience into three interrelated dimensions: 1) structural robustness, concerning the network's fundamental capacity to maintain connectivity after an attack (aligned with percolation theory); 2) strategic protection efficacy, pertaining to the identification and safeguarding of nodes critical to network functionality (corresponding to optimal network disintegration theory); and 3) dynamic stability, involving the ability to suppress the propagation of local failures into global collapse (related to cascading failure theory). By integrating key node identification with metrics of robustness, protection, and stability into a composite index, this paper operationalizes these theoretical dimensions into a hybrid resilience metric—quantified as the survivability rate, the critical node protection rate, and the cascade stability. Ultimately, this approach provides a multidimensional characterization of a network's response to attacks or cascading failures, establishing a more robust scientific foundation for resilience analysis and optimization.

In conclusion, there are still several research gaps in critical node identification and resilience evaluation: (1) Critical node identification lacks unified frameworks integrating multi-dimensional attributes (e.g., topology, dynamic failure impact), relying mainly on static metrics or isolated ML features with limited interpretability and generalizability [11,12]. (2) The assessment of infrastructure network resilience is limited by the use of isolated metrics (e.g., survival rate, node protection, cascade suppression) [13–15], which fail to account for the multidimensional nature of resilience and thus offer only partial insight into system robustness and recovery. These limitations constrain reliable network optimization and risk mitigation against cascading failures. To bridge these research gaps, this paper models infrastructure networks as undirected weighted graphs. A TEC-GNN model is proposed for the identification of critical nodes, which exploits features from both static (topological structure, entropy) and dynamic (cascading impact) dimensions. The model integrates nodal topological centrality measures, such as degree centrality, closeness centrality, and betweenness centrality, along with information entropy, and the influence of cascading failures, which is measured by the scale of failures triggered by a node. Based on the critical node ranking generated by the TEC-GNN, a comprehensive resilience metric is developed, incorporating survival rate, critical node protection, and cascading stability. Furthermore, a real-world case study verifies the effectiveness of the framework.

The remainder of this paper is organized as follows: Section 2 provides a literature review of cascading failure mechanisms, critical node identification and resilience analysis. Section 3 provides the methodology. Section 4 discusses the simulation results, and finally, section 5 concludes this study. The overall research workflow is depicted in Fig 1.

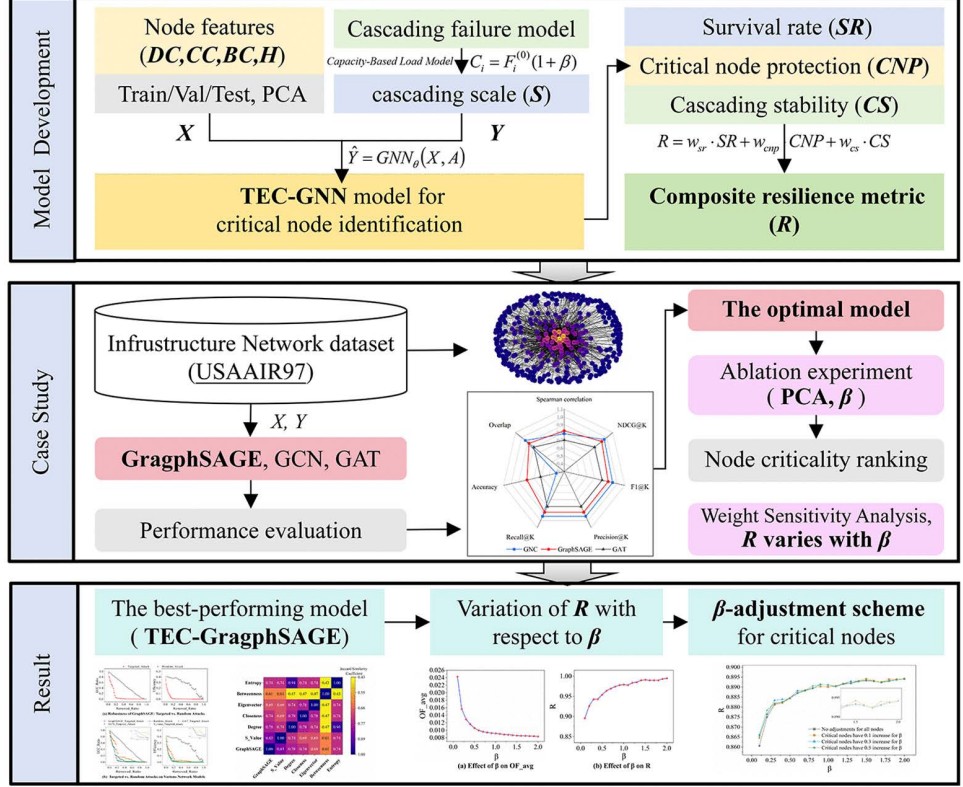

**Fig 1. Overall research workflow of this paper.**

## Literature review

Early research predominantly explored cascading failures in power grids and communication networks. By leveraging simple network models (e.g., node-edge structures), basic propagation mechanisms were put forward, and cascade modeling was utilized to analyze fault diffusion pathways [17–19]. With the progress of complex network theory, studies introduced various models, such as small-world and scale-free networks. These models systematically expounded the influence mechanisms of topological features (e.g., degree distribution, clustering coefficient) on the critical thresholds for cascading failures and their propagation scope [20–23]. More recently, simulation techniques have been widely employed to quantify the temporal evolution of failures and identify critical nodes within the systems. As a result, failure mitigation strategies based on topology optimization or capacity allocation have been proposed [24,25].

In network theory, a critical node refers to one whose failure or removal leads to a substantial deterioration in the network's performance, connectivity, or robustness. These nodes generally occupy central positions within the network. Consequently, the detection and preservation of critical nodes are of utmost importance for maintaining the overall stability and functional integrity of the system. Artime et al. classify existing critical node identification methods into two main types [16]. The first type is based on static topological information, leveraging structural features such as degree and betweenness centrality to evaluate node importance. The second type focuses on dynamic information flow, analyzing processes like information propagation to identify key nodes. Topological centrality metrics offer fundamental evaluations by quantifying the local influence of nodes. This includes degree centrality, which measures node connectivity; betweenness centrality, which assesses control over information flow; and closeness centrality, which evaluates path efficiency [26–29].

Topological information, such as network structural position analysis, reveals hierarchical organization through methods like the k-shell decomposition proposed by Yang et al. [30] to examine the positional importance of nodes within networks. Propagation dynamics models, such as the SIR/SI frameworks, are widely employed to simulate information or disease diffusion and quantify the propagation capabilities of nodes [29]. Meanwhile, machine learning and optimization algorithms facilitate data-driven predictions by training on topological features to assess node importance [31]. In summary, existing node criticality assessment methods exhibit significant limitations by relying too heavily on network topology and ignoring the effects of cascading failures on network resilience.

The development of critical node identification methods has driven corresponding progress in network resilience research. Resilience assessment can identify topological vulnerabilities, quantify the risks of cascading failures, and guide protective strategies (e.g., redundancy design) to improve robustness. Resilience analysis in complex networks is fundamentally based on two methodological paradigms: network dismantling and optimal percolation [16]. Network dismantling aims to disrupt connectivity by removing critical nodes or edges, and has given rise to various approaches. These include topology-based attacks using centrality metrics such as degree and betweenness [32,33], dynamic optimization algorithms like Collective Influence, which iteratively minimizes the leading eigenvalue of the non-backtracking matrix and machine learning techniques, especially Graph Neural Networks as demonstrated by the GDM algorithm [34]. In contrast, optimal percolation is focused on identifying the minimum set of nodes whose removal can fragment the giant connected component [35]. Furthermore, existing studies establish resilience evaluation frameworks from two perspectives: structural resilience (topological properties) and functional resilience (cascading stability). The structural aspect emphasizes topological robustness through metrics such as the size of the largest connected component [36], network efficiency [37], and node connectivity [28]. The functional resilience focuses on evaluating post-failure performance through service availability and traffic rerouting rate [38]. Recent trends involve integrating multi-metric approaches and developing composite indicators that incorporate survival rates and hybrid metrics [39,40]. For example, Chen et al. proposed a resilience assessment metric based on the proportion of node failures, quantifying the critical threshold of network collapse through percolation theory to reveal the nonlinear relationship between the tolerance parameter and the failure scale [35]. Subsequently, the Qi research team developed an indicator that incorporates critical node protection, validating the functional recovery capabilities of infrastructure networks through simulations [41]. However, existing studies mainly adopt single methodological approaches, and the integrated analysis considering both structural and functional resilience is a crucial area for further research.

From the above literature review, it is evident that existing studies on critical node identification have predominantly progressed along two distinct paths: static topological analysis and dynamic failure modeling. Nevertheless, these methods have mostly evolved independently, achieving limited success in integrating structural metrics and dynamic nodal impacts into a unified evaluation framework. Furthermore, infrastructure resilience assessment is still restricted by fragmented evaluation metrics (e.g., survival rate [39], cascade suppression [35,41]), which fail to comprehensively capture system robustness and recovery dynamics. To address these crucial gaps, this study puts forward an integrated framework that combines static topological analysis with dynamic failure modeling for critical node identification. The framework systematically evaluates node criticality from three dimensions: structural position, functional role under stress conditions, and potential to trigger cascading failures. By bridging the methodological gap between purely structural metrics and isolated dynamic assessments, this approach offers a multidimensional characterization of node importance, providing a more robust and comprehensive foundation for resilience-oriented network analysis and decision-making.

In this research, infrastructure stations and lines are modeled as an undirected, weighted network. We construct a two-dimensional Graph Neural Network (GNN) model using a composite resilience metric. The main contributions are: (1) the TEC-GNN framework, integrating PCA-reduced topological centralities and cascade-impact signals to jointly capture the structural and dynamic functional roles of nodes, surpassing prior reliance on static topology; (2) a composite

resilience metric that unifies survival rate, critical node protection, and cascade stability for holistic network degradation assessment; (3) validation in a real-world air transportation network, with accurate identification of critical nodes, demonstration of the nonlinear influence of redundancy coefficient $\beta$ on resilience, and cost-effectiveness of targeted $\beta$-tuning.

This research presents a methodological framework that integrates critical node identification with resilience evaluation. The proposed approach equips infrastructure operators across transportation, power systems, and urban management with actionable insights. It enables the development of targeted protection strategies and increases robustness against cascading failures. The approach directly aids infrastructure emergency management and preventive maintenance planning.

## Methodology

### Mechanisms of cascading failure propagation

Complex networks, such as infrastructure networks, demonstrate structural intricacy and a high degree of functional interdependence, rendering them susceptible to cascading failures instigated by localized disruptions. The functional reliability of infrastructure networks, including transportation and power systems, is highly contingent upon critical node facilities, such as transportation hubs and substations [3–5].These facilities, serving as core functional aggregation units, assume essential responsibilities such as resource transfer and processing. Their failure not only directly disrupts network connectivity but also propagates the impact of faults. In contrast, edge failures generally lead only to a localized decline in transmission efficiency. Empirical incidents, such as the 2021 Suez Canal obstruction causing regional supply-chain disruptions and multiple large-scale blackouts in California, USA, which are often triggered by critical node failures, highlight the substantial risk posed by node failures [2,7]. Therefore, this paper primarily explores cascading failures in infrastructure networks under node failure scenarios. When a node fails, it can prompt the cessation of operations in upstream components while simultaneously disabling downstream functions. This dual-disruption mechanism propagates through the network, ultimately causing systemic supply-demand imbalances. The process of cascading failure subsequent to a node failure is depicted in Fig 2.

Within the network topology, transportation hubs and substations are modeled as nodes, while resource flows and transfers are conceptualized as edges. We model the infrastructure network as a weighted undirected graph, $G=\{V, E\}$, where $V=(1, 2,..., n)$ is a set of nodes and $A=\{(i, j)|a_{ij}=1 \text{ or } 0\}$ is a set of edges. Here, $e_{ij}$ denotes the connectivity between node $i$ and node $j$.

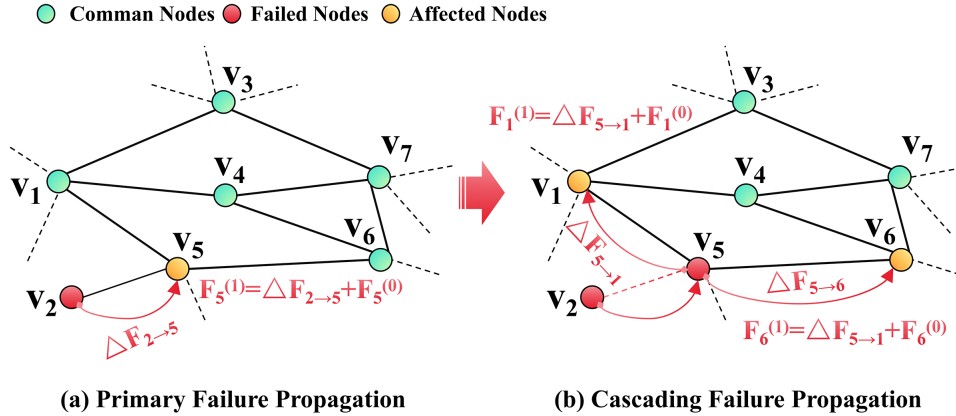

(a) Primary Failure Propagation (b) Cascading Failure Propagation

**Fig 2. An example of the cascading failure propagation process.**

In the proposed infrastructure networks, each node $i$ possesses an initial load $F_i^{(0)}$ and a load capacity threshold $C_i$, where $F_i$ denotes the operational demand at equilibrium state and $C_i$ specifies the failure threshold. Based on centrality metrics such as degree, numerous models have been developed to characterize the functional relationship between capacity threshold $C_i$ and initial load $F_i^{(0)}$. In this paper, we utilize the model proposed by Motter and Lai [42], where each node is assigned the maximum load it can support (Eq. (1)).

$$C_i = F_i^{(0)}(1 + \beta) \tag{1}$$

Where $i = 1, 2,..., n$, $\beta$ is the redundancy coefficient of the network, obviously $\beta > 0$.

In infrastructure networks, such as air-transport systems, eigenvector centrality ($EC$) can effectively reflect the criticality of nodes by taking into account both direct connections and the influence of neighboring nodes [43]. An airport that is connected to major hubs (e.g., Beijing or Shanghai) assumes a crucial role, even when its own traffic volume is relatively low, as it functions as a gateway to core nodes. By accounting for the "quality of neighbors" effect, eigenvector centrality can effectively reflect potential cascade risks and systemic vulnerability, providing a more realistic assessment of node criticality compared to degree centrality alone. In this study, $EC$ is designated as the initial load for each node. To simulate real-world dynamics, $EC$ values are iteratively updated throughout the cascade process, continuously mirroring the structural changes subsequent to each failure event.

As Fig 2 shows, when node $i$ ($i = 1,2, ..., n$) fails, the load it carries is typically redistributed among adjacent nodes $j$ within the network. For instance, in Fig 1, when node $V_2$ fails, its load is reallocated to the adjacent node $V_5$. Then the current load of $V_5$ is updated from $F_5^{(0)}$ to $F_5^{(1)}$ ($F_5^{(1)} = \Delta F_{2 \to 5} + F_5$), where $\Delta F_{2 \to 5}$ is the load of node $V_2$ redistributed to $V_5$. Once $F_5^{(1)} > C_5$, the overload $F_5^{(1)} - C_5$ at node $V_5$ will then be reallocated to its adjacent nodes $V_1$ and $V_6$ based on a pre-determined allocation rule, and node $V_5$ transitions to a failed state due to capacity overload. If node $V_1$ and node $V_6$ can handle the overload from node $V_5$, only a small portion of the network will fail (i.e., node $V_2$ and $V_5$, edge $e_{2,5}$), otherwise, the failure may affect other neighboring nodes, and the propagation of the cascading failure continues until no additional node fails due to overload. The impact of cascading failure propagates through the network, potentially terminating at a specific node or resulting in the failure of all nodes within the network, finally leading to a large-scale collapse or complete paralysis of the whole system. Therefore, the reallocated load from failed nodes is critical to the propagation of cascading failures throughout the network.

Drawing upon the cascading failure model put forward by Chen [35], the overload of a failed node can be re-distributed to its adjacent nodes in accordance with the connectivity strength. In infrastructure networks, edge weights act as quantitative indicators of the strength of nodal connections, taking into account physical constraints like transport distance and line capacity. Load distribution strategies founded on these weights facilitate the efficient identification of critical paths and nodes, thereby augmenting network resilience and alleviating cascading failures. Consequently, in this paper, the load redistribution strategy is closely associated with the edge weight that quantifies nodal association(Eq. (2)).

$$\Delta F_{i \to j} = \begin{cases} F_i \cdot \dfrac{w_{ij}}{\sum\limits_{k} w_{ik}} & \text{where i is a first failed node} \\[4mm] (F_i - C_i) \cdot \dfrac{w_{ij}}{\sum\limits_{k} w_{ik}} & \text{where i is a cascading failed node} \end{cases} \tag{2}$$

where, $\Delta F_{i \to j}$ represents the extra load allocated to node $V_j$ from node $V_i$, and, $F_i$ denotes the current load of node $V_i$ prior to the load redistribution triggered by failed nodes, $w_{ij}$ represents the edge weight characterizing the connection strength

between node $V_i$ and its direct neighbor $V_j$, $k$ denotes the number of adjacent nodes of node $V_i$, and $w_{ik}$ denotes the sum of all the connection strength between node $V_i$ and its direct neighbor $V_j$.

Therefore, there exists two adjacency matrices: $A_{n \times n} = \{a_{ij}\}_{n \times n}$, $W_{n \times n} = \{w_{ij}\}_{n \times n}$.

Based on node failures and load redistribution mechanisms, the schematic of the cascading failure process under node failure scenarios is illustrated in Fig 3. Each iteration of load redistribution necessitates the updating of node information (failure/non-failure), the edge connectivity status $e_{ij}$, the temporary load of each node $F_i^{(t)} = EC_i^{(t)}$, where $t$ represents the number of iterations and $t = 1, 2, ..., n$.

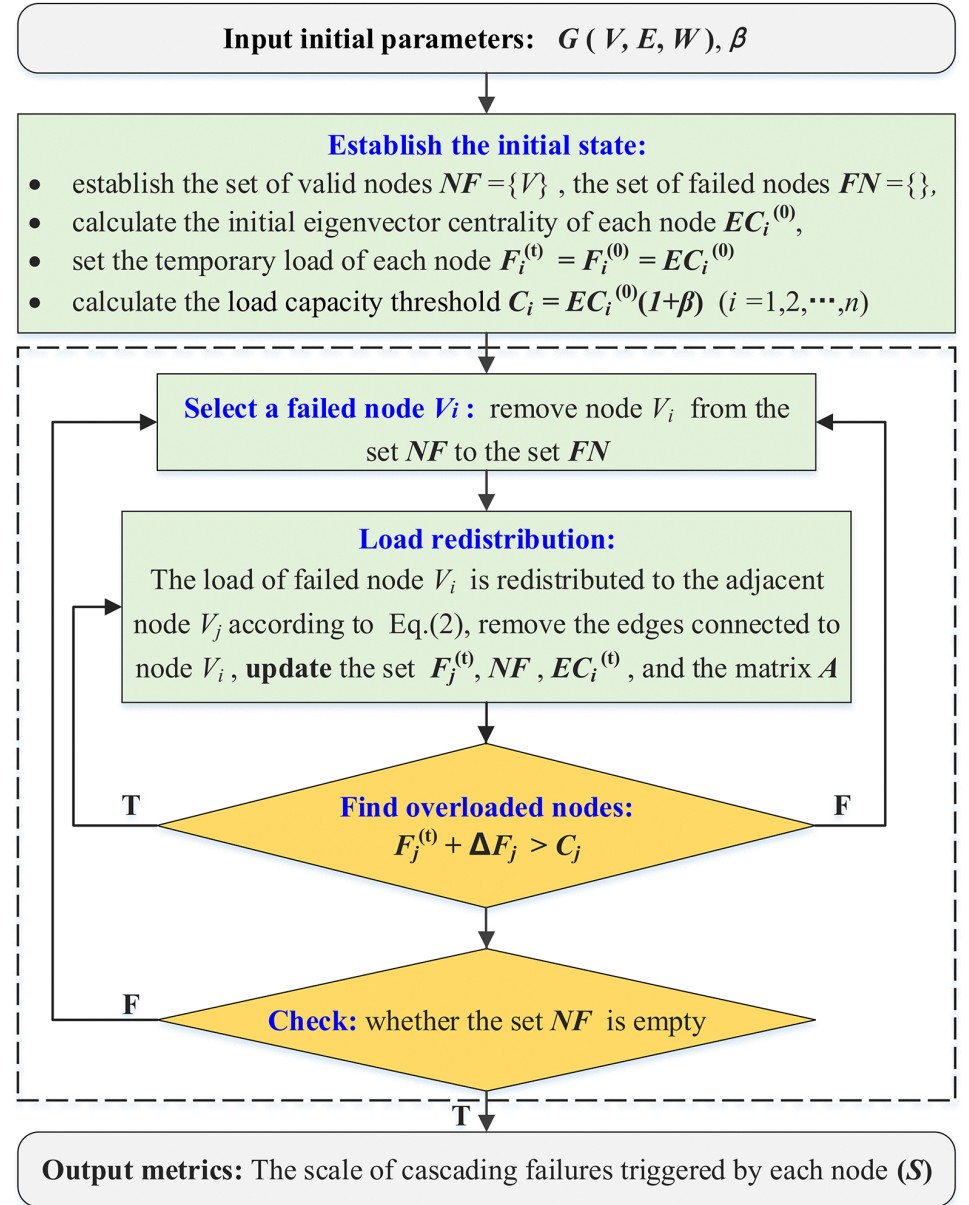

**Fig 3. Schematic of the cascading failure process.**

## TEC-GNN for critical node identification

The well-established approach for identifying critical nodes involves utilizing metrics to quantify the structural roles, functional influence, and vulnerability of nodes. These metrics predominantly encompass degree centrality($DC$), betweenness centrality($BC$), closeness centrality($CC$), and eigenvector centrality($EC$), and they exhibit a high degree of correlation with the network topological structure [26, 27]. Node information entropy measures the uncertainty or variety of information around a network node. It is based on principles from information theory. To calculate it, one uses the logarithmic entropy of the probability distribution of states among neighboring nodes. Researchers often use this metric to identify important nodes in complex networks. For example, Wu et al. [44] suggested a k-order entropy method to spot influential nodes. Huang et al. [45] used adjacency-based entropy to identify key nodes in urban rail transit networks. Maji [46] introduced Directional Node Strength Entropy (DNSE) for identifying critical nodes in undirected networks. Degree centrality simply counts a node's connections. In contrast, node information entropy looks at variation in a node's neighborhood information. Nodes with high entropy have diverse connections and complex information flows. These nodes often act as network hubs. This method combines the quality and diversity of connections using more precise mathematical methods. As a result, it can better distinguish node roles and more accurately find nodes crucial for network stability and function. Recently, researchers have started using graph neural networks based on these entropy ideas. This approach helps overcome some of the limitations of older structural metrics. In contrast to traditional methods that rely on predefined rules (such as single or composite centrality indices), TEC-GNN (Topology-Entropy-Cascading Graph Neural Network) automatically learn to fuse multi-scale topological information via nonlinear message passing. This provides a more advanced, data-driven assessment of a node's criticality for network robustness and resilience.

Therefore, in this paper, three GNN models (GCN, GraphSAGE, and GAT) are respectively employed to identify critical nodes. To tackle the potential high linear correlations among graph centrality metrics ($DC, CC, BC, H$), which may undermine the model accuracy, Principal Component Analysis (PCA) is applied for dimensionality reduction as a pre-processing step to enhance the training performance. The input feature matrix ($X$) for each model consists of the principal components (e.g., $P_1$, $P_2$) obtained through PCA, and the scale of cascading failures ($S$) is used as the supervisory signal ($Y$) for training. Based on these, a research architecture diagram is presented as Fig 4.

As Fig 4 shows, a multidimensional critical nodes identification framework is established. The proposed framework leverages a supervised learning paradigm to identify critical nodes. Following the computation of centrality-based features ($X$), three GNN models are trained using the cascading failure scale ($S$) as the supervisory signal. A comparative evaluation of the models identifies the top performer, which is then deployed to assign a criticality score to each node, pinpointing those most likely to cause large-scale failures.

Following the framework described above, the TEC-GNN model is developed to identify critical nodes within networks as Eq.(3–6).

$$\hat{Y} = GNN_\theta(X, A) \tag{3}$$

$$\theta^* = \arg\min_\theta L\left(\hat{Y}, Y\right) + \lambda\|\theta\|_2 \tag{4}$$

$$X = [x_1, x_2, \cdots, x_n]^T \in R^{n \times 4}, x_i = [P1(i), P2(i), \cdots] \tag{5}$$

$$Y = [S(1), S(2), \cdots, S(n)]^T \in R^n, S(i) = CascadeScale(V_i, G) \tag{6}$$

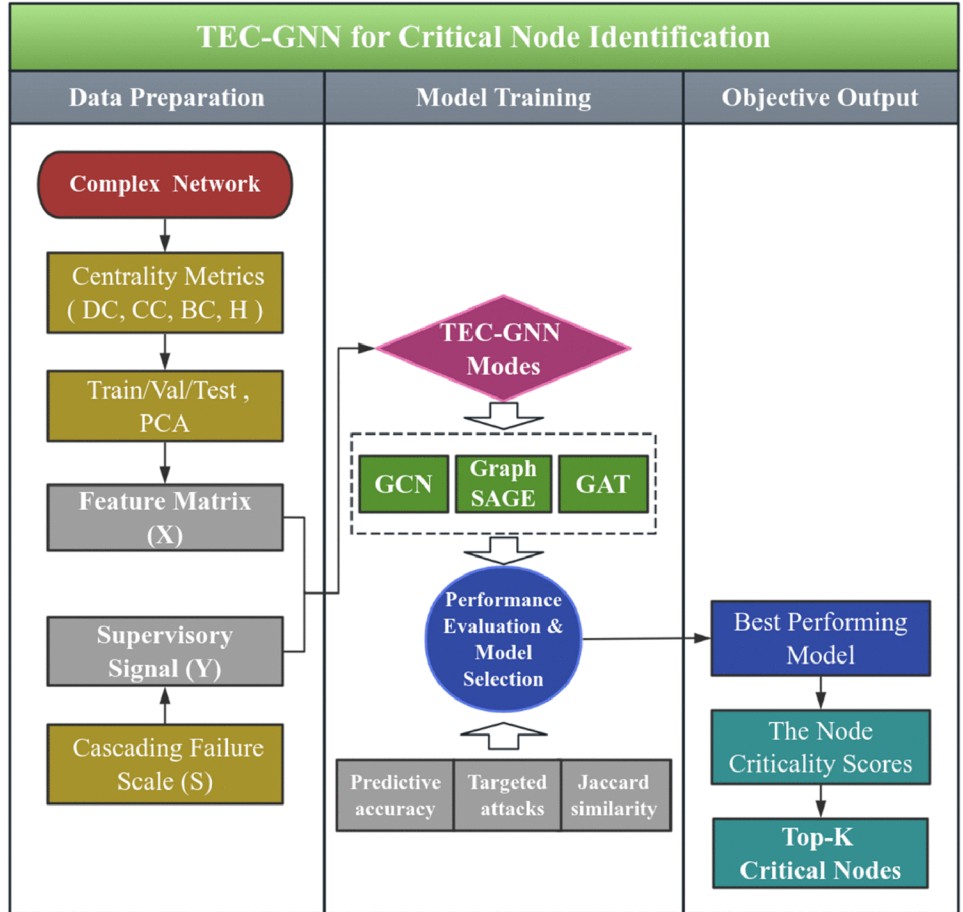

**Fig 4. Flowchart of critical nodes identification process.**

where $\theta$ denotes the trainable parameters of the TEC-GNN model, $A$ represents the adjacency matrix encoding the graph structure, $A_{ij} = 1$ if nodes $V_i$ and $V_j$ are connected, and 0 otherwise ($i, j = 1, 2,..., n$). $\hat{Y} \in R^n$ is the predicted criticality scores for all nodes in the network, higher values indicate greater node importance. $Y \in R^n$ is the ground-truth supervisory signal vector containing cascading failure impact scores $S(i$ for each node, $\lambda$ denotes the regularization coefficient that controls the penalty term $\|\theta\|_2$. $L((Y, \hat{Y})$ is the loss function that quantifies the discrepancy between predicted and actual criticality scores. Eq. (4) shows the optimization objective for identifying the optimal model parameters $\theta*$ that minimize the combined loss of prediction error and model complexity through L2 regularization.

The feature set ($X$) is constructed by implementing principal component analysis (PCA) for dimensionality reduction on the four metrics (*degree centrality (DC), closeness centrality (CC), betweenness centrality (BC), and node entropy (H)*), with the resultant principal components functioning as the input features ($P1, P2,...$). The four metrics are respectively formulated in accordance with Eq. (7–11). *DC* reflects the immediate influence through the number of its connections, *CC* indicates the global efficiency by means of the average shortest – path distance, and *BC* represents the strategic brokerage by counting the shortest paths it enables. Node Entropy ($H$) quantifies the disorder in a node's edge weights. The cascading failure scale ($S$) quantifies the magnitude of the impact when node $V_i$ fails.

$$DC(i) = \sum_{j=1}^{n} A_{ij} = \deg(i) \tag{7}$$

$$CC(i) = \frac{n-1}{\sum_{j=1}^{n} d_{(i,j)}} \tag{8}$$

$$BC(i) = \sum_{s \neq t \neq i} \frac{\delta_{st}(i)}{\delta_{st}} \tag{9}$$

$$H(i) = -\sum_{j \in N_i} p_{ij} \log_2(p_{ij}) \tag{10}$$

$$S(i) = \frac{\textit{Number of cascading failure nodes triggered by node } V_i}{n} \tag{11}$$

Where: *A* represents the adjacency matrix, deg(*i*) is the number of edges connected to node $V_i$, *n* is the total number of nodes in the network, $d_{(i,j)}$ represents the shortest-path distance between nodes $V_i$ and $V_j$, $\delta_{st}$ is the total number of shortest paths from *s* to *t*, $\delta_{st}(i)$ is the number of those paths passing through $V_i$. N(i) is a set of neighbors of node $V_i$, $w_{ij}$ is weight of edge $e_{i,j}$.

In the process of identifying critical nodes through TEC-GNN, the centrality-information-based feature vectors (*degree centrality (DC), closeness centrality (CC), betweenness centrality (BC), and Entropy (H)*) present a multi-dimensional depiction of topological significance. This enables the model to comprehensively capture both the local and global structural functions of nodes. The supervised signal, defined by the cascading failure scale (*S*), provides a direct and influential metric for assessing node criticality. It guides the model to transcend the mere learning of structural attributes and delve into the actual functional vulnerability.

By integrating these features, the TEC-GNN learns to map intricate topological patterns to practical failure impacts, thereby alleviating the limitations of single-metric methods. The model's capacity to generalize from the combined feature space and the supervisory signal enhances its robustness in the face of noisy or incomplete structural data. This data-driven approach not only automates the identification process but also captures the nonlinear interactions among centrality metrics. Consequently, it establishes a more accurate and adaptable evaluation framework that is suitable for diverse network structures and failure scenarios.

### Composite indicator development for resilience assessment

Resilience refers to the capacity of a network to maintain functionality in the face of node failures or attacks, or more comprehensively, the ability of a system to recover from or adapt to disturbances [47]. Contemporary assessment methods for the resilience of complex networks are mainly classified into static and dynamic approaches. Static methods assess the robustness of a network against failures by employing topological metrics (e.g., average path length and clustering coefficient). A typical example is the connectivity analysis under node/edge removal put forward by Djelloul et al. [48]. In contrast, dynamic methods concentrate on the propagation of disturbances and the system recovery processes. Prominent examples encompass the cascading failure model introduced by Pan et al. and the temporal evolution framework

developed by Cumming et al. [49,50]. Hybrid approaches combine both static and dynamic perspectives. For example, Tan et al. proposed the "robust yet fragile" theory, which quantifies the difference in network resilience under localized attacks compared to global disturbances [51].

In this study, a hybrid assessment method is adopted, which incorporates the survival rate, critical node protection, and cascade stability into a comprehensive resilience indicator (Eq. (12–15)). The Survival Rate (*SR*) quantifies the overall network resilience under a specific fault intensity, with a higher value signifying a stronger ability to uphold functional integrity. The Critical Node Protection rate (*CNP*) assesses the efficacy of protecting the most crucial nodes, mirroring the performance of targeted protection strategies. Finally, Cascade Stability (*CS*) gauges the predictability of failure propagation, where an increase in value indicates better failure suppression capabilities.

Within this framework, the protection of critical nodes is executed based on the top-K critical nodes identified in Fig 4.

$$R = w_{sr} \cdot SR + w_{cnp} \cdot CNP + w_{cs} \cdot CS \tag{12}$$

$$SR_{\beta} = 1 - OF_{\beta} \tag{13}$$

$$CNP_{\beta} = \frac{1}{|T|} \sum_{i \in T} \left(1 - \frac{f_i}{M}\right) \tag{14}$$

$$CS_{\beta} = \frac{1/\sigma_{\beta}}{\max_{\beta' \in B} (1/\sigma_{\beta'})} \tag{15}$$

where $\beta \in B$ is the redundancy coefficient of the network, $OF_{\beta} = N_{failed} / N_{total}$ represents the overall failure proportion of the system under redundancy coefficient $\beta$. $N_{failed}$ denotes the total number of failed nodes, and $N_{total}$ is the total number of nodes in the network. *T* is the set of top-K critical nodes identified by the GNN model. *M* is the total number of simulations, $f_i$ is the number of failures of node $V_i$ over *M* simulations. $\sigma_{\beta}$ is the standard deviation of cascade steps under $\beta$. *B* is the set of all considered redundancy coefficient $\beta$. The terms $w_{sr}$, $w_{cnp}$, $w_{cs}$ represent the weights for the three indicators.

The hybrid network resilience assessment framework developed in this study systematically measures the multi-dimensional resilience attributes proposed by Artime et al. [16] through three metrics: survival rate, critical node protection rate, and cascade stability. Among these, the survival rate anchors the structural robustness dimension, using the core order parameter of percolation theory (the relative size of the maximum connected component) to quantify a network's structural survival capability at specific fault intensities. The critical node protection rate breaks from traditional node importance metrics by employing GraphSAGE to identify key nodes, aligning with the cutting-edge direction of "leveraging machine learning to discern higher-order topological patterns for precise network disintegration." This achieves a deep integration of methodology with network disintegration theory. Cascade stability addresses the dynamic stability dimension by quantifying the system's ability to suppress cascade collapse via the variance in fault propagation in simulations, enabling more refined measurement of dynamic failure processes. These three metrics anchor structural, strategic, and dynamic theoretical dimensions, respectively. Through methods such as seepage computation, graph neural networks, and dynamic variance analysis, they form an empirical closed loop, constructing a nested "method-metric-theory" system. This approach overcomes single-dimensional limitations while encompassing performance outcomes, structural criticality, and dynamic failure processes, providing a multidimensional, process-oriented reference framework for system resilience assessment.

   

## Simulations and results

The aim of our simulation research is two-pronged: (1) to ascertain the most efficacious TEC-GNN model for the identification of critical nodes within infrastructure networks; (2) to analyze the influence of the parameter $\beta$ on the resilience metric $R$. We carry out a comparative assessment of multiple GNN architectures (e.g., GCN, GAT, GraphSAGE) to achieve the first objective. Regarding the second one, we investigate how fluctuations in $\beta$ affect the $R$-value under the cascading failure scenario and the effect of targeted $\beta$ adjustments at critical nodes on network resilience.

### Experimental setup

This section expounds on the fundamental components of our simulation experiments, encompassing the datasets utilized, the implementation particulars of our proposed model, the baseline methods for comparison, and the evaluation metrics adopted to gauge performance.

### Dataset description and PCA preprocessing

The performance of the proposed model was evaluated on the publicly accessible benchmark datasets. Specifically, USAir97, an infrastructure network depicting the 1997 US air transportation system, is a weighted network consisting of 332 airports (nodes) and 2,126 direct flight routes (edges), with the latter weighted by flight frequency [52]. Table 1 presents a summary of the basic statistics of the dataset, and Fig 5 showcases the network topology, the distributions of graph centrality metrics ($DC$, $CC$, $BC$, $H$), and the Pearson correlation coefficient among the four metrics.

The distributions of the four centrality metrics ($DC$, $CC$, $BC$, $H$), computed from the network data and shown in Fig 5 (S1 Table in S1 File: Corresponding data for centrality metrics of all nodes), and Fig 5(b) demonstrates high correlations among them. Given the high inter-metric correlations (e.g., Pearson correlation: $CC$ vs. $BC = 0.84$) and associated redundancy, PCA was thus used for dimensionality reduction, condensing the data into a smaller set of principal components that retain essential information. With a cumulative variance contribution rate of 92.8%, the first two principal components (PCs) sufficiently capture the information of the original four variables. The suitability for PCA is supported by a KMO value of 0.76 and a significant Bartlett's test ($p < 0.001$), justifying their extraction as integrated features. These PCs form a two-dimensional feature vector, $X = [P1(i), P2(i)]$, for each node, which is used as input to the subsequent GNN model.

The Total Variance Explained table indicates that the first principal component (P1) exhibits approximately moderate positive loadings (0.43–0.54) across all variables, suggesting that it represents a "comprehensive centrality" measure reflecting a node's overall influence and connection strength in the network. In contrast, the second principal component (P2) displays a clear structural differentiation: $BC$ shows a strong positive loading (0.83), while $CC$ and Entropy $H$ exhibit notable negative loading ($-0.37$, $-0.40$). This loading pattern reveals a dual classification of node criticality: nodes with high P2 scores act as "bridges" connecting different network modules, whereas those with low P2 scores function as "core hubs" characterized by high integration and propagation efficiency. Capturing the distinct dimensions of "connection strength" (P1) and "structural role" (P2), the two mutually orthogonal PCs thus provide a more robust and comprehensive feature vector for identifying critical nodes (Table 2).

**Table 1. Statistical summary of USAir97.**

| Dataset | |V| | |E| | <k> | D | C | r |
|---|---|---|---|---|---|---|
| **USAIR97** | 332 | 2126 | 12 | 0.038692 | 0.625217 | −0.207876 |

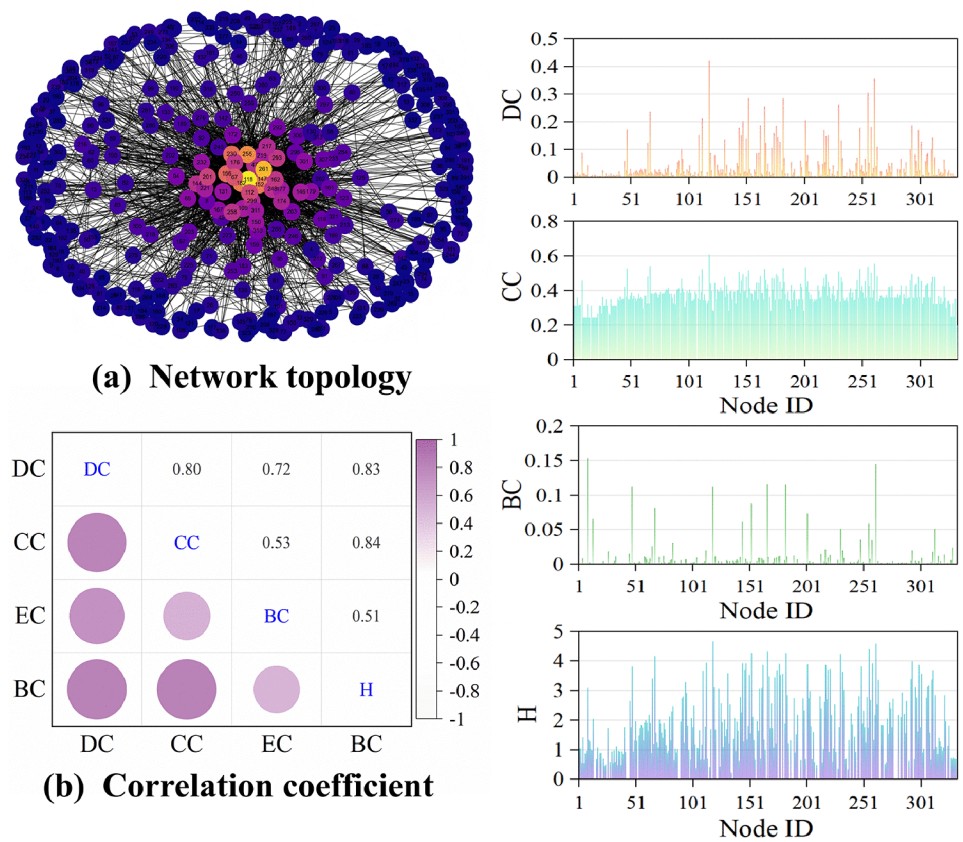

**(a) Network topology**

**(b) Correlation coefficient**

**Fig 5. Basic topological information with centrality measures.**

**Table 2. Total variance explained.**

| Original Variables | P1 | P2 |
|---|---|---|
| DC | 0.54 | 0.06 |
| CC | 0.51 | −0.37 |
| BC | 0.43 | 0.83 |
| H | 0.51 | −0.40 |

## Implementation details and hyperparameters

We conduct a comparison of our approach with three representative graph neural networks: the Graph Convolutional Network (GCN) for standard convolutional operations, GraphSAGE which employs mean aggregation for inductive learning, and the Graph Attention Network (GAT) equipped with multi − head attention mechanisms. Our model was implemented utilizing the PyTorch Geometric library. The model was trained within a CPU − based environment (Intel Core i5-1135G7 @ 2.40GHz) under the following configuration. To ensure a fair comparison, all models were trained with the same data splits (Train/Val/Test: 60%/20%/20%), optimization algorithms (the Adam optimizer with a learning rate of 0.005), and evaluation protocols (Table S1 in S1 File). Model-specific hyperparameters were individually adjusted to attain optimal performance while preserving the architectural integrity, detailed parameter settings are provided in Table S2 in S1 File.

## Evaluation metrics

In the task of critical node identification, the selection of evaluation metrics follows a multifaceted framework that thoroughly assesses model performance across distinct yet complementary dimensions (Table S4 in S1 File). Effect Size is measured using Cohen's d, which quantifies the standardized difference between critical and ordinary nodes. This ensures the model captures meaningful differences. Ranking Consistency is evaluated via Spearman's ρ (with confidence intervals) to check the stability and reliability of node ordering. Ranking Quality uses both Spearman correlation and position-sensitive metrics such as NDCG@K (Normalized Discounted Cumulative Gain at Top-K) and F1@K (F1 Score at Top-K: Evaluating the harmonic mean of precision and recall). This balances overall rank correlation with accuracy in top-K predictions. Top-K Identification Accuracy is assessed using Precision@K, Recall@K, and overall Accuracy. These metrics provide insights into the model's exact classification performance on the most important nodes. Ranking Agreement (Overlap) compares consistency with alternative ranking methods. Computational Efficiency is measured by inference time, ensuring practicality for deployment. Together, these metrics provide a holistic evaluation of a model's ability to reliably identify, rank, and classify critical nodes under real-world constraints.

## Performance evaluation of critical node identification

Based on the critical node identification methodology depicted in Fig 4, the performance is assessed from four complementary dimensions: (1) Predictive accuracy of the GAT, GNC, and GraphSAGE models (Fig 6, S1 File; Table 3, S4 in S1 File). (2) Ablation studies on PCA and the $\beta$-parameter to assess feature contribution and fusion sensitivity (Fig 7). (3) Network robustness under targeted and random attacks (Fig 8). (4) Ranking consistency via Jaccard similarity with centrality-based rankings (Fig 9). These ablation studies provide deeper insights into the functional importance of specific model components, ensuring the identified critical nodes are not only accurate but also methodologically interpretable.

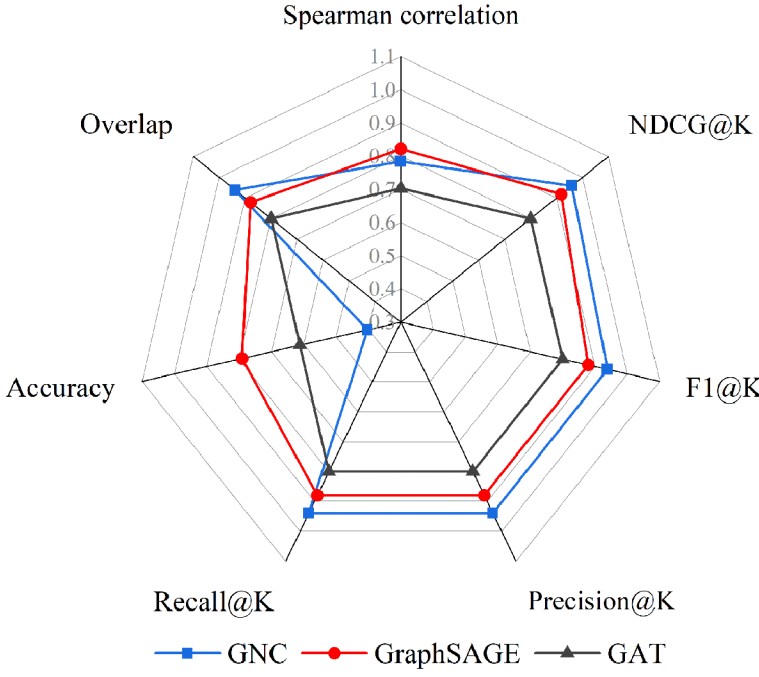

**Fig 6. Performance evaluation on critical node identification.**

**Table 3. The performance evaluation results: Cohen's d, Inference time and Spearman correlation.**

| Metrics | GNC | GraphSAGE | GAT |
|---|---|---|---|
| Cohen's d | 4.703 | 2.814 | 2.561 |
| Inference time | 0.002s | 0.002s | 0.006s |
| Spearman correlation | 0.784 | 0.822 | 0.703 |
| Spearman's ρ (95% CI) | [0.738, 0.825] | [0.777, 0.855] | [0.643, 0.757] |

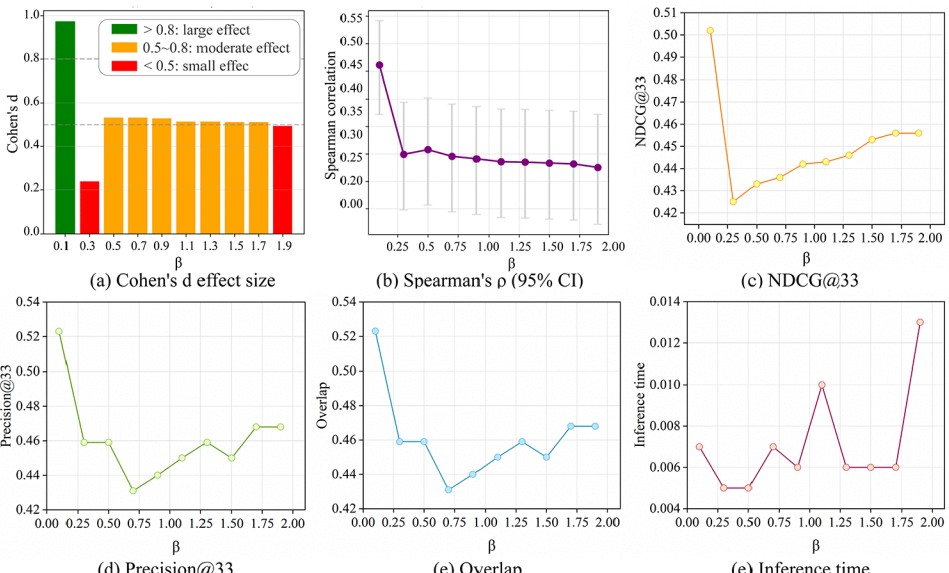

(a) Cohen's d effect size  (b) Spearman's ρ (95% CI)  (c) NDCG@33
(d) Precision@33  (e) Overlap  (e) Inference time

**Fig 7. Changes in evaluation metrics of the GraphSAGE model under _β_ -ablation experiments.**

In accordance with the evaluation metrics defined for the three GNN models, the performance evaluation results of the three models obtained from the simulation experiments are presented in Fig. 6, S1, and Table 3, S4. Based on the comprehensive evaluation metrics, GraphSAGE demonstrates a balanced and robust advantage in critical node identification among the three GNN models. Quantitatively, it achieves strong performance across multiple dimensions: in ranking consistency, it attains a Spearman correlation of 0.822 with a tight confidence interval [0.777, 0.855]; in ranking quality, it records high NDCG@K (0.918) and F1@K (0.879); and in top-K identification, it maintains Precision@K and Recall@K both at 0.879. Although its effect size (Cohen's d = 2.814) is slightly lower than GNC, it remains well above the large-effect threshold (≥0.8). Qualitatively, GraphSAGE offers a superior trade-off between predictive accuracy and computational efficiency; its inference time (0.002s) matches that of GNC and is significantly faster than GAT, while still delivering competitive and stable ranking results. This combination of consistent accuracy, interpretable ranking behavior, and low computational cost positions GraphSAGE as the most practical and well-balanced model for scalable critical node identification.

Table 4 compares GraphSAGE performance using original versus PCA-reduced features. PCA preprocessing significantly improves multiple key metrics while maintaining stability in others. Cohen's d improved from 2.814 to 2.928 (Δ + 0.113, + 4.03%). NDCG@K increased from 0.918 to 0.959 (Δ + 0.041, + 4.43%). Classification metrics showed robust gains: F1@K, Precision@K, Recall@K, and Overlap each rose from 0.879 to 0.939 (Δ + 0.061, + 6.90%). Accuracy improved from 0.791 to 0.821 (Δ + 0.030, + 3.78%). Spearman's correlation stayed nearly unchanged at 0.821 (Δ = 0.001, −0.17%), and inference time remained stable at 0.002 seconds. These results indicate that PCA strengthens model

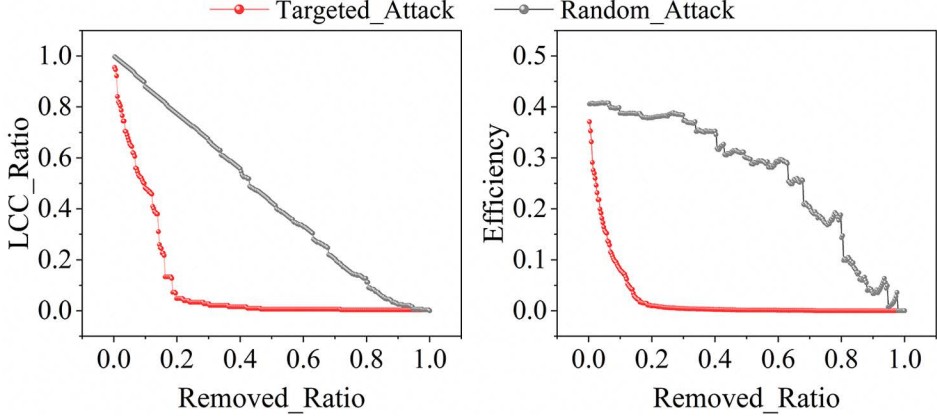

**(a) Robustness of GraphSAGE: Targeted vs. Random Attacks**

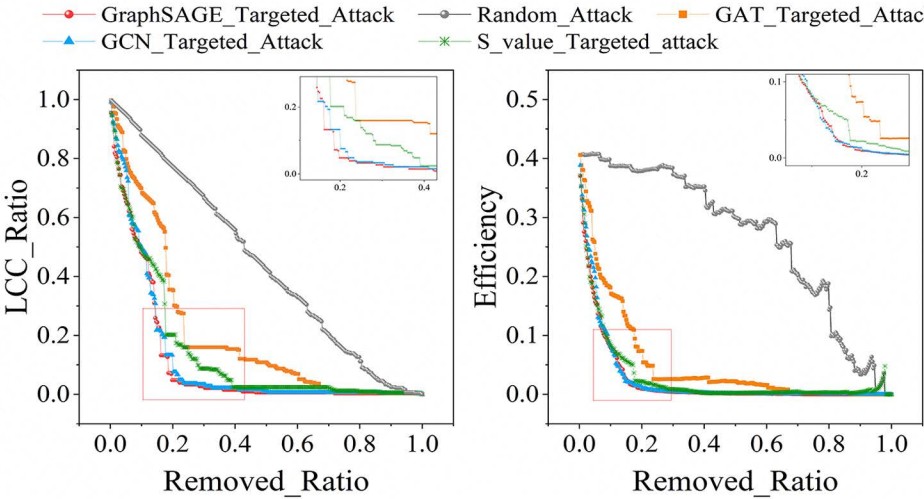

**(b) Targeted vs. Random Attacks on Various Network Models**

**Fig 8. Targeted vs. Random Attacks: GraphSAGE versus baseline methods.**

discriminative and ranking performance without compromising efficiency or correlation, supporting its utility for feature refinement in graph-based learning.

The β-ablation experiments on the GraphSAGE model show its multifaceted response to the β hyperparameter, as visualized in Fig 7 and S5 Table in S1 File. Cohen's d effect size (Fig 7a) shows a large effect at $\beta = 0.1$ (green). It shows small effects at $\beta = 0.3$ and $\beta = 1.9$ (red). The effects are predominantly moderate (orange) across intermediate values (0.5–1.7). This demonstrates β's varying influence on performance significance. For ranking metrics, Spearman's ρ (Fig 7b) declines sharply between $\beta = 0.25$ and 0.5 before stabilizing. This suggests an initial disruption in ranking consistency. NDCG@33 (Fig 7c) drops notably at $\beta = 0.25$, but partially recovers with higher β. Precision@33 (Fig 7d) fluctuates, reaching a minimum around $\beta = 0.75$. This highlights sensitivity to β-driven precision-ranking trade-offs. Feature/label overlap (Fig 7e) decreases from $\beta = 0.0$ to 0.5, reaches a trough at $\beta = 0.75$, and rebounds slightly at higher β. This suggests shifts in alignment between learned representations and target signals. Inference time (Fig 7f) decreases up to $\beta = 0.25$, then surges at $\beta = 0.5$ and peaks at $\beta = 2.0$. This reflects complexity-efficiency trade-offs. These results show β's complex, non-linear

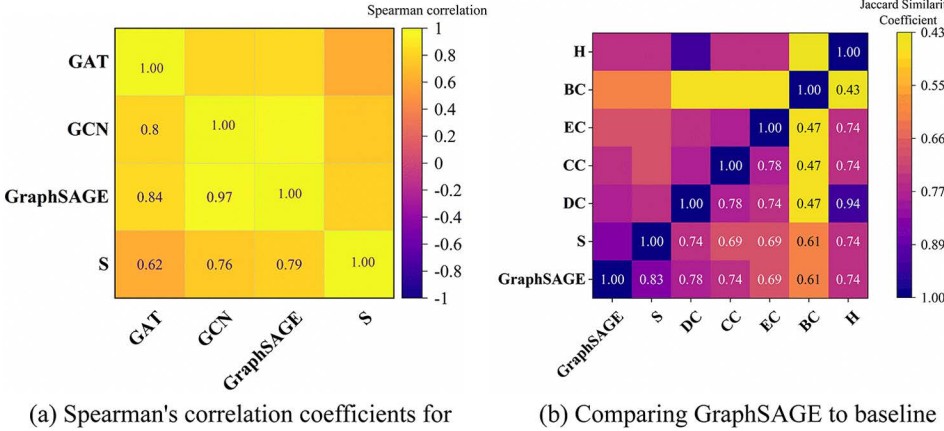

(a) Spearman's correlation coefficients for node ranking across three GNN models

(b) Comparing GraphSAGE to baseline methods for node ranking

**Fig 9. Comparing GraphSAGE to baseline methods for node ranking.**

**Table 4. Changes in evaluation metrics of the GraphSAGE model: comparison of input feature vectors before and after PCA ablation.**

| Metric | Original | PCA | Abs_Change | Rel_Change(%) | Conclusion |
|---|---|---|---|---|---|
| Cohen's d | 2.814 | 2.928 | 0.113 | ↑ 4.03% | Improvement |
| Spearman's ρ (95% CI) | [0.777, 0.855] | [0.780, 0.854] | No significant change | No significant change | Minimal variation |
| Spearman correlation | 0.822 | 0.821 | −0.001 | ↓ −0.17% | Minimal variation |
| NDCG@K | 0.918 | 0.959 | 0.041 | ↑ 4.43% | Improvement |
| F1@K | 0.879 | 0.939 | 0.061 | ↑ 6.90% | Improvement |
| Precision@K | 0.879 | 0.939 | 0.061 | ↑ 6.90% | Improvement |
| Recall@K | 0.879 | 0.939 | 0.061 | ↑ 6.90% | Improvement |
| Accuracy | 0.791 | 0.821 | 0.030 | ↑ 3.78% | Improvement |
| Overlap | 0.8788 | 0.939 | 0.060 | ↑ 6.89% | Improvement |
| Inference time | 0.002s | 0.002s | No significant change | No significant change | Minimal variation |

impact on significance, ranking efficacy, representational alignment, and efficiency. They emphasize the need for balanced hyperparameter tuning.

We assessed network robustness using two key metrics: the Largest Connected Component (LCC) Ratio and Network Efficiency. Simulations were performed under two attack scenarios: (a) Targeted Attacks, involving deliberate removal of the top 10% most critical nodes, and (b) Random Attacks, in which nodes failed at random. In the targeted attack, nodes were removed in descending order of their criticality scores, starting with the highest-ranked node (Tables S6 and S7 in S1 File). Fig 8(a) shows that attacking critical nodes identified by GraphSAGE results in a sharp decline of LCC_Ratio to about 0.4 when 10% of nodes are removed. This suggests near-complete network fragmentation. At a 20% removal ratio, the LCC ratio drops below 0.2. In contrast, under random attack, the network maintains an LCC_Ratio of around 0.4 even after 60% of nodes are removed. This significant gap (20% vs. 60%) highlights the infrastructure network's vulnerability due to its heavy reliance on a few critical nodes. Fig 8(b) further shows that the decline in LCC_Ratio and Efficiency under targeted attacks occurs faster for GraphSAGE than for GCN and GAT. This demonstrates greater accuracy in identifying key nodes for network connectivity. The results validate GraphSAGE's effectiveness. It can identify high-impact nodes

precisely and significantly enhance attack efficiency. These findings confirm that GraphSAGE is a strong tool for evaluating and improving the robustness and resilience of infrastructure networks.

As illustrated in Fig 9 (Tables S8 and S9 in S1 File: corresponding to the list data for Fig 9(a) and 9(b), which present critical node rankings in list format, comparing GraphSAGE with baseline methods, thereby enabling granular validation of the conclusions derived from Fig 9.), GraphSAGE demonstrates superior performance in critical node ranking. In Fig 9(a), the matrix of Spearman correlation coefficients shows that GraphSAGE achieves a higher correlation with the supervisory signal $S$ (0.79). This is greater than GCN (0.76) and GAT (0.62) and reflects stronger alignment with the ground-truth ranking objective. This advantage is further corroborated in Fig 9 (b), where GraphSAGE attains the highest Jaccard similarity (0.83) with the actual impact of node failures $S$. This value substantially exceeds its similarity with any individual centrality metric (≤0.74). Moreover, GraphSAGE shows moderate similarity with classical centrality measures such as degree and betweenness centrality (ranging from 0.47 to 0.78). This suggests that the model effectively captures structural information that conventional methods often overlook. Collectively, these results confirm that GraphSAGE not only learns the supervised ranking target more accurately but also identifies hub nodes whose removal critically disrupts network connectivity. The model thus exhibits comprehensive and robust performance in critical node identification.

## Effects of *β* and targeted tuning on resilience

The exploration of the impact of the redundancy coefficient $β$ on the resilience indicator $R$ holds significant importance for the security and stability of critical infrastructure networks, such as air transportation systems, which are intrinsically susceptible to cascading failures induced by localized disruptions. Analyzing $R(β)$ uncovers the fundamental trade-off between the redundancy cost and the resultant level of operational resilience. Based on the network resilience indicator calculation model presented in Eq. (12–15), this section investigates the dependence of the resilience indicator $R$ on the redundancy coefficient $β$ (where $β$ ranges from 0.1 to 2.0 with an increment of 0.1) by utilizing the GraphSAGE-based critical node identification model.

As defined in Eq. (12), the parameters ($w_{sr}$, $w_{cnp}$, and $w_{cs}$) represent weights for survival rate, critical node protection, and cascading stability. We use fixed weights ($w_{sr}$ = 0.5, $w_{cnp}$ = 0.3, $w_{cs}$ = 0.2) to build the evaluation framework. This allocation stems from theoretical consensus and empirical findings in infrastructure resilience research. It aims to balance system functionality, node importance, and dynamic risk. First, the highest weight (0.5) goes to survivability. This reflects resilience's main goal: preserving core system services. Most studies use survivability as the primary metric and target [53,54]. The critical node protection weight (0.3) shows a key trade-off in network robustness. Over-fortifying a few hubs may help against targeted attacks, but it can also raise vulnerability to multiple failures. This trade-off is widely supported in resource-constrained defense strategy research [35,55]. The cascade stability weight (0.2) matches its roughly 1:2 co-survival ratio and the proportions of cascade losses observed in real events. Studies show that in networks such as power grids and transportation systems, cascading effects account for about 20–30% of losses. Targeting these effects increases resilience by about the same share [47,56]. Overall, this system covers core resilience concerns—structure, function, and dynamics—offering both consistency and practical guidance.

To test the weighting scheme, we first check how the weights fluctuate. Fig 10(a) shows that variation differs for each component: $w_{sr}$ (0.47–0.53, mean: 0.5), $w_{cnp}$ (0.27–0.33, mean: 0.3), and $w_{cs}$ (0.16–0.24, mean: 0.2) over 200 samples (see Table S10 in S1 File). This variation reflects each component's sensitivity to parameter changes. It highlights the need for empirical weight allocation to avoid overfitting. We further examine the link between weight shifts and the resilience metric $R$. Fig 10(b) gives Spearman's correlation between weight changes (via $β$) and $R$. The correlation stays near zero with no trend across all tested values. Thus, moderate shifts from fixed weights do not alter $R$ in one direction, proving the allocation's stability. Sensitivity analysis backs up the 2:1 ratio for $w_{sr}$ to $w_{cs}$. Fig 10(c) shows R's path under baseline and altered weights. The paths are closely aligned, so $R$ stays in a narrow range even with weight changes. In Fig 10(d), relative $R$ changes always stay under 1.5%. Error bars remain small, showing low variability. This insensitivity

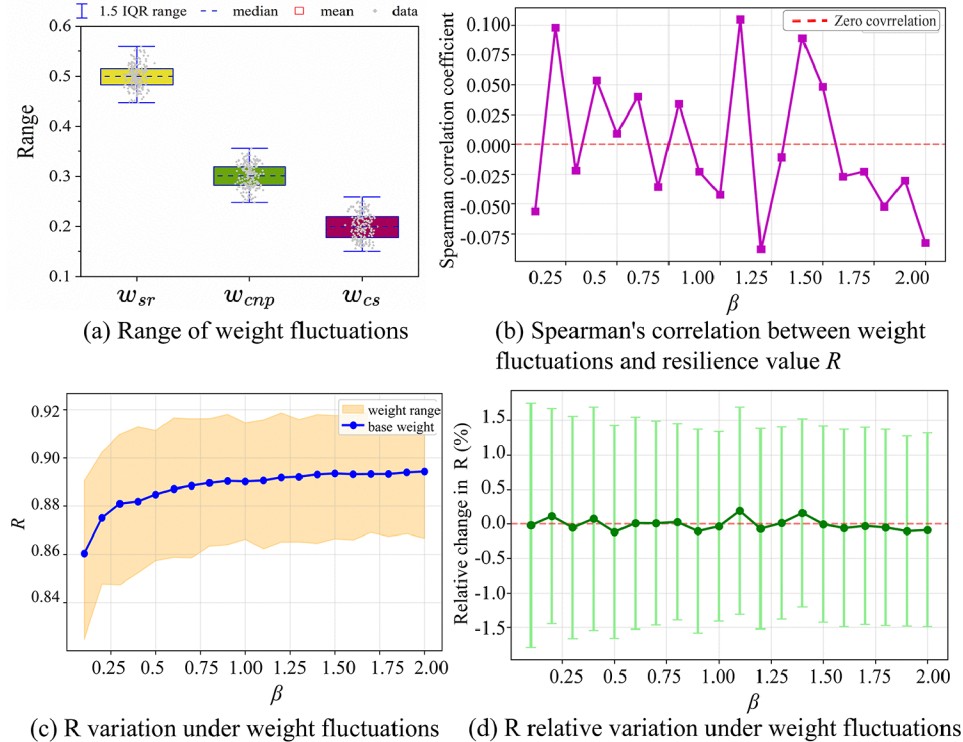

(a) Range of weight fluctuations

(b) Spearman's correlation between weight fluctuations and resilience value $R$

(c) R variation under weight fluctuations

(d) R relative variation under weight fluctuations

**Fig 10. Sensitivity analysis of resilience metric weights.**

means resilience rankings remain stable, which is vital for comparison. In summary, graphical and statistical checks confirm that the weighting is stable and reliable for assessing resilience in complex systems such as air transport networks.

### The fundamental impact of $\beta$ on cascade-based resilience $R$

Fig 11(a) illustrates the effect of the redundancy coefficient $\beta$ on the average cascade failure ratio $OF\_avg$ within the network. As $\beta$ increases, $OF\_avg$ decreases significantly before gradually levelling off. Specifically, when $\beta$ rises from 0.0 to 0.3, $OF\_avg$ drops rapidly by 38.8%. Beyond this point, further increasing $\beta$ to the experimental maximum of 2.0 leads to a markedly slower reduction. These results suggest that network redundancy can substantially improve robustness in the early stage, but beyond a threshold around $\beta \approx 0.3$, the enhancement exhibits clear diminishing marginal returns.

Fig 11(b) presents the relationship between $\beta$ and the network resilience indicator $R$. As $\beta$ increases from 0.0 to 2.0, $R$ rises initially and then stabilizes. In the range of $\beta = 0.0$ to 0.5, $R$ grows rapidly. Beyond $\beta = 0.5$, further increases result only in minor fluctuations, with $R$ eventually approaching 1.0. This indicates that while moderate redundancy effectively strengthens resilience, exceeding $\beta \approx 0.5$ leads to saturation in improvement, again reflecting diminishing marginal gains.

Notably, $R$ and $OF\_avg$ show a strong negative correlation as $\beta$ varies: while $R$ increases, $OF\_avg$ decreases. This inverse relationship highlights a fundamental trade-off in infrastructure network design. For real-world systems such as air transportation systems, where resources are limited, this trade-off underscores the importance of selecting an optimal $\beta$ that balances resilience enhancement with cost efficiency. The results provide a quantitative basis for determining redundancy strategies that maximize robustness while avoiding over-investment in diminishing returns.

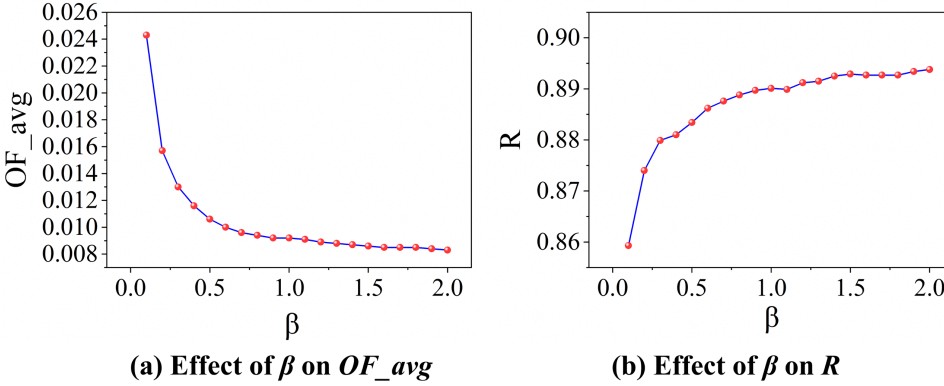

(a) Effect of *β* on *OF_avg*        (b) Effect of *β* on *R*

**Fig 11.  Effect of *β* on cascade failure ratio and network resilience.**

## Enhancing resilience via targeted *β*-tuning

Fig 12 shows that targeted adjustment of the redundancy coefficient $\beta$ at critical nodes enhances network resilience $R$. Compared to the baseline, increasing $\beta$ at these nodes consistently improves $R$. For instance, at a global $\beta$ of 0.2, raising $\beta$ by 0.3 at critical nodes increases $R$ from 0.874 to 0.883. As the global $\beta$ rises, $R$ values increase in all cases, and strategies focusing on critical nodes, especially with increments of 0.3 and 0.5, have an advantage. These results confirm that targeted redundancy allocation at critical nodes is effective for resilience optimization. By increasing $\beta$ at hubs identified by methods like GraphSAGE, cascading failure propagation can be mitigated without system-wide redundancy upgrades. This approach uses the nonlinear efficacy of redundancy, concentrating resources for maximum robustness gains, achieving significant resilience improvements with minimal investment and offering an economically sustainable strategy for critical infrastructure networks.

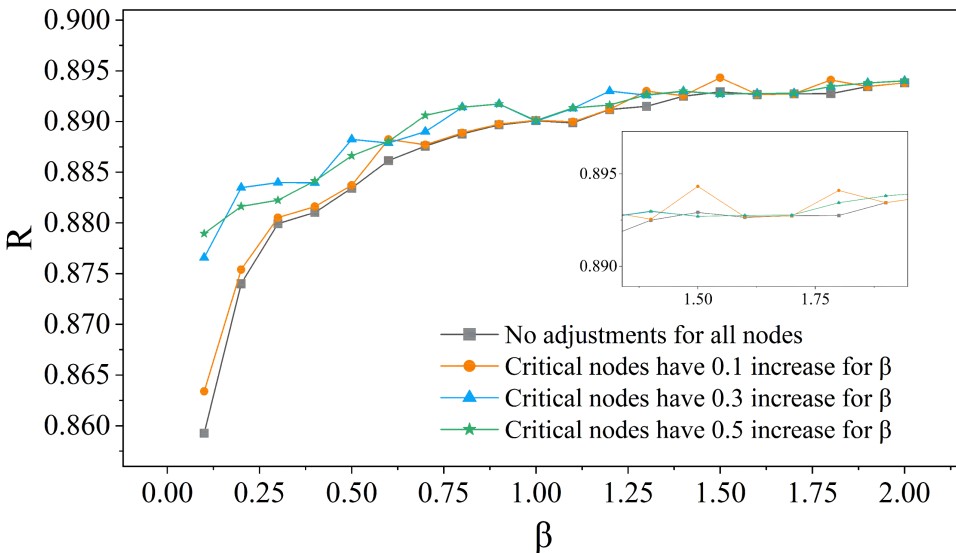

**Fig 12.  Targeted β tuning for improved resilience at critical nodes.**

The optimal adjustment magnitude for critical node's $\beta$ values varies with different intervals, related to the system's structural and dynamic characteristics in different $\beta$ ranges. In the low $\beta$ range (0.25–0.5), a moderate 0.3 adjustment can activate critical-node synergy for optimal resilience. In the medium $\beta$ range (0.5–1.5), a 0.5 adjustment is needed to disrupt equilibrium and unlock greater resilience. When $\beta$ is in the higher range (1.5–1.75), a 0.1 increase can optimize resilience through precise resource allocation and interaction efficiency regulation. This interval-specific behavior provides a basis for tailored resilience enhancement strategies in practical engineering based on the system's initial $\beta$ value.

## Scalability and large-scale network applications

This study evaluates computational efficiency and scalability through both algorithm design and experimental validation. Results show that the proposed TEC-GraphSAGE framework remains resilient and efficient for processing large-scale networks (Supplementary Material: S14 in S1 File), successfully handling networks with up to 10,000 nodes on a single machine and indicating robustness for even larger scales. The following sections discuss three dimensions: computational complexity, practical performance, and scalability.

Computational complexity centers on three stages: multi-indicator centrality and node entropy computation, cascade failure simulation, and GraphSAGE graph neural network training. During feature computation, the framework selects degree centrality, closeness centrality, betweenness centrality, and node entropy as node input features. Node entropy is the sum of the edge weights connected to the node. Feature vector centrality is used only for node load allocation in the cascading failure model; it does not participate in feature construction. Degree centrality is the most efficient to compute. Closeness centrality requires full-graph shortest path calculations. Exact betweenness centrality has a complexity of $O(nm)$. For large networks, we use the Brandes approximation algorithm, which is based on k-node sampling. This balances accuracy and efficiency. Cascade failure simulation is intrinsically parallel and has a time complexity of $O(k \cdot n \cdot t)$. By sampling different load parameters $\beta$ and running simulations independently, we can accelerate this process using parallel computation. For graph neural network training, our study uses the GraphSAGE model, which has a per-layer time complexity of $O(n \cdot d2 + m)$. In contrast to models like Graph Attention Networks (GAT), which compute attention weights for all node pairs, GraphSAGE uses fixed neighbor sampling and aggregation. This method maintains high performance while reducing memory and computational overhead. As a result, it is suitable for inductive learning on large-scale graph data.

To validate the method's practical efficiency, we conducted a comprehensive empirical analysis of the USAAir97 network (a medium-sized network) comprising 332 nodes. The entire process took approximately 20 minutes, distributed as follows: the one-time computation of four input features (degree, proximity, betweenness centrality, and node entropy) took about 15 minutes; training the GraphSAGE model took approximately 2 minutes; the remainder was spent on statistical analysis of the results. It is crucial to note that the most computationally intensive component lies in the dynamic simulation of cascading failures. This process requires initiating k·n independent simulations for k· $\beta$ values and n nodes, with each iteration dynamically recalculating the network-wide feature vector centrality to allocate load. The cumulative computational complexity of this process significantly exceeds that of static feature extraction. This implementation employs parallelization strategies to accelerate this process. Based on the above analysis, the theoretical scalability of this method is clearly established. For networks with node sizes ranging from 1,000–10,000, computations can be expected to complete within hours on a single machine equipped with a standard GPU. For large networks with more than 10,000 nodes, a distributed computing framework is recommended to accelerate the cascading simulation. This framework's implementation is entirely based on PyTorch Geometric, a specialized graph neural network library. Its built-in efficient sparse computation, neighbor sampling, and GPU acceleration mechanisms provide the underlying support for processing larger-scale networks. For ultra-large networks (nodes>100,000), further optimization of memory and computational efficiency at extreme scales can be achieved by leveraging advanced features of this framework (e.g., subgraph training) or by integrating with other frameworks, such as the Deep Graph Library (DGL).

This framework can scale to multi-layer network architectures through three technical approaches. First, mapping multi-layer networks as hypergraphs allows direct modeling of cross-layer higher-order interactions using hypergraph neural networks. Second, a layered processing strategy evaluates resilience independently across layers, followed by global synthesis using coupling matrices and fusion functions. Third, constructing three-dimensional adjacency tensor representations lays the groundwork for developing native multi-layer graph convolution operators. These approaches enable more precise characterization of complex coupling relationships in real-world systems but also increase computational and modeling complexity. Current work still faces limitations in memory handling for ultra-large-scale networks. Future research will focus on developing distributed simulation algorithms, graph compression training techniques, and online learning mechanisms for dynamic networks to further enhance the framework's applicability and practicality.

## Discussions

The robustness and resilience of critical infrastructure networks, including aviation, power grids, and communication systems, are key system security concerns [1–3,5]. Accurate identification of critical nodes and targeted reinforcement strategies is essential for preventing cascading failures and improving system resilience [13–15]. Graph neural networks (GNNs) have recently shown promise in this domain due to their advanced graph structure learning capabilities [11,12,34]. This study systematically evaluates several GNN models for critical node identification. It also examines the nonlinear regulatory effects of the redundancy coefficient $\beta$ on the resilience of complex networks. Results show that GraphSAGE outperforms other models on key metrics. Its Spearman correlation coefficient with the supervised signal S is 0.79 (GCN: 0.76; GAT: 0.62). Its NDCG@K (0.918) and F1@K (0.879) reflect strong ranking quality and balance. In Top-K identification, both Precision@K and Recall@K reach 0.879, which indicates high detection accuracy. This performance stems from GraphSAGE's inductive learning and fixed-size neighbor sampling. These features offer robust generalization for dynamic networks. Inference time is just 0.002 seconds, matching GCN and surpassing GAT. The effect size (Cohen's d = 2.814) further confirms the statistical significance of its high accuracy and efficiency. This meets the performance-cost needs of large-scale network analysis. GraphSAGE offers advantages in critical node identification that align with the operational needs of large-scale infrastructure networks for dynamic analysis and rapid response. Research by Artime et al.(2024) [16] supports that GraphSAGE's inductive neighborhood aggregation mechanism enables stable and general identification of key hubs within the topology. This is particularly true in strategic protection, compared to GAT models that depend on global attention. Precise identification forms a reliable foundation for later network reinforcement and resilience strategies. This includes calculating metrics such as survival rate and cascade stability.

Network robustness experiments provide direct validation of model efficacy. Targeted attacks on the top 10% of critical nodes identified by GraphSAGE demonstrate that removing only 10% of nodes reduces the network's largest connected component ratio (LCC_Ratio). Network robustness experiments directly validate the efficacy of the proposed models. Targeted attacks on the top 10% of critical nodes identified by GraphSAGE show that removing just 10% of nodes reduces the network's largest connected component ratio (LCC_Ratio) to approximately 0.4, effectively paralyzing the network. Increasing the removal rate to 20% causes the LCC_Ratio to drop below 0.2. In contrast, random attacks require removing up to 60% of nodes to achieve a similar level of disruption. This significant difference between 20% targeted and 60% random attacks quantitatively demonstrates the high dependency of infrastructure networks on a small subset of structural hub nodes. It also provides inverse validation of GraphSAGE's precision in identifying critical nodes, confirming that the identified nodes represent core vulnerabilities in network resilience. This result is consistent with Gao et al.'s [47] conclusion that network vulnerability arises from node heterogeneity. Furthermore, it advances the quantifiable validation of hub node identification, offering empirical support for resilience enhancement strategies focused on critical node reinforcement.d) increased from 2.814 to 2.928 (a relative increase of 4.03%), and ranking quality (NDCG@K) improved from 0.918 to 0.959 (a relative increase of 4.43%); Classification performance showed even more significant improvement, with F1@K, Precision@K, and Recall@K all increased from 0.879 to 0.939, with all three metrics increasing by 6.90%.

This enhancement stems from PCA's effective removal of redundant and noisy features in the original data, enabling the distillation of discriminative information and thereby strengthening the model's core recognition capabilities. Notably, the dimensionality reduction process did not compromise other critical model properties—Spearman correlation (0.822 vs. 0.821) and inference time (0.002s) remained stable. This provides empirical evidence for graph learning that "scientific dimensionality reduction can enhance model performance without sacrificing key information."

The redundancy coefficient $\beta$, as the core parameter linking model identification and network resilience optimization, exhibits a distinctly nonlinear influence mechanism. $\beta$-ablation experiments reveal: the effect size (Cohen's d) peaks at $\beta = 0.1$ (large effect) and drops to its minimum at $\beta = 0.3$ and $\beta = 1.9$ (small effect); Rank consistency (Spearman's $\rho$) declines sharply between $\beta = 0.25$ and 0.5 before stabilizing; inference time exhibits non-monotonic variation, surging abnormally at $\beta = 0.5$. This indicates that $\beta$ not only modulates statistical significance but simultaneously impacts ranking quality, feature-label alignment efficiency, and computational complexity. Practical applications require targeted optimization to adapt to specific network architectures, theoretically aligning with Chen et al. (2022)'s [35] research on parameter sensitivity in network robustness. Further analysis of $\beta$'s effect on resilience metric R reveals diminishing marginal returns: R increases rapidly as $\beta$ rises from 0.0 to 0.5, then slows and fluctuates slightly beyond 0.5. Concurrently, the average cascade failure rate (OF_avg) exhibits a strong negative correlation with R. This quantitative relationship reveals the "cost-benefit" trade-off in redundancy investment: moderately increasing $\beta$ effectively enhances resilience, but marginal benefits decline significantly beyond the threshold. This provides quantitative evidence for formulating optimal redundancy strategies in resource-constrained infrastructure systems such as aviation networks.

This study reveals a "precision reinforcement" strategy for allocating redundant resources. By increasing the global redundancy coefficient $\beta$ by 0.3 only for critical nodes identified by GraphSAGE, while keeping $\beta = 0.2$ for the rest of the network, network resilience R rises from 0.874 to 0.883. This advantage remains stable across different global $\beta$ ranges. This result confirms that concentrating resources on critical nodes can provide a nonlinear boost to resilience at low cost. System robustness improves because the propagation of cascading failures is suppressed. This provides a practical, cost-effective path for engineering: instead of costly full-network upgrades, targeted reinforcement of key nodes efficiently boosts robustness. The findings support the idea that precision investment is better than blind expansion in resource-constrained situations.

The algorithmic design in this study prioritizes scalability. Theoretical analysis indicates that computational complexity primarily stems from two modules: cascade failure simulation ($O(k \cdot n \cdot t)$) and GraphSAGE training ($O(n \cdot d^2 + m)$). Through combined technical optimizations, complexity is effectively controlled: the Brandes approximation algorithm reduces cascading simulation overhead, while leveraging GraphSAGE's inherent neighbor sampling mechanism and parallel computing framework accelerates GNN training. Empirical results demonstrate that on the 332-node USAAir97 network, the entire process takes approximately 20 minutes, with the cascading simulation portion further compressible through parallelization. This indicates that for networks with 1,000–10,000 nodes, the method can complete within hours on a single GPU. For larger networks (>10,000 nodes), scalability can be achieved by leveraging the distributed capabilities of frameworks like PyTorch Geometric. However, the current framework faces memory constraints for ultra-large networks (>100,000 nodes), necessitating the integration of subgraph training and online learning techniques to enhance its applicability.

This study has three limitations. First, simulations of large-scale cascading failures require significant computational resources. Second, the network topology is assumed to be static, which does not fully reflect real network dynamics, such as new node additions or changes in edge weights. Third, node features are based on traditional centrality metrics, without using multimodal data such as real-time traffic or device status. Also, how well the framework scales to multi-layer coupled networks needs more testing. Future research can focus on four areas: 1) Develop distributed cascading simulation algorithms and dynamic graph learning for real-time analysis and online decisions. This would ease computational pressures and adapt to network dynamics. 2) Improve models by adding spatio-temporal graph neural networks

or knowledge graphs. This would integrate temporal information and domain knowledge. 3) Expand architecture by using hypergraph neural networks or tensor decomposition to handle higher-order interactions in multi-layer networks. 4) Deepen application by applying the framework to other infrastructure, like smart grids or supply chain networks. Validate and refine with real-world events to move theory toward practice.

## Conclusions

Accurately identifying critical nodes and optimizing network resilience are key scientific challenges for secure, stable critical infrastructure. This study builds a comprehensive analytical framework (TEC-GNN) that combines graph neural networks, feature engineering, and resilience assessments. It systematically checks the performance of different models for finding critical nodes. It also explains regulatory mechanisms and paths for improving network redundancy and resilience. Key findings are: 1) The GraphSAGE model performs best at spotting critical nodes, with a high Spearman correlation (0.822), NDCG@K (0.918), F1@K (0.879), and Top-K accuracy (0.879). Its inference efficiency (0.002s) matches GNC and is much faster than GAT. 2) PCA dimensionality reduction boosts the model's ability to distinguish between nodes and improves ranking, with Cohen's d up by 4.03%, NDCG@K up by 4.43%, and classification metrics improved by about 6.90%. These gains do not sacrifice ranking consistency or efficiency. 3) The redundancy coefficient $\beta$ strongly and nonlinearly affects the model's effect size, ranking consistency, and efficiency. Careful tuning is needed to find the best $\beta$. 4) Removing 20% of the critical nodes (as identified by GraphSAGE) seriously damages the network, three times more effectively than a random attack. This confirms the structural importance of identified nodes. 5) As $\beta$ increases, network resilience metric R shows diminishing returns. Reinforcing $\beta$ for critical nodes—"precision reinforcement"—achieves large improvements in resilience at low cost, making it a cost-effective strategy.

This study proposes an interpretable, scalable framework for identifying critical nodes and assessing resilience. It deepens the use of graph neural networks for analyzing the reliability of complex systems. In practice, these findings support risk assessment, vulnerability protection, and resource optimization in critical infrastructure sectors such as transport, energy, and communications. The "precision reinforcement" strategy is well-suited for improving robustness when resources are limited. The research mainly uses static network designs. Adapting to dynamic and multi-layer networks still needs work. Large cascading simulations consume significant computing power. Feature construction could add more data sources. Future studies should target efficient dynamic graph learning, distributed simulation, and multi-layer interdependent networks. They should also focus on adding real-time data for online monitoring and adaptive optimization of infrastructure resilience.

## Supporting information

**S1 File. Supplementary Material.**
(PDF)

## Author contributions

**Conceptualization:** Anqi Liu, Wenfu Zhao.

**Formal analysis:** Wenfu Zhao.

**Investigation:** Wenfu Zhao.

**Methodology:** Anqi Liu, Wenfu Zhao.

**Project administration:** Anqi Liu.

**Resources:** Anqi Liu.

**Software:** Wenfu Zhao.

**Supervision:** Anqi Liu.

**Visualization:** Wenfu Zhao.

**Writing – original draft:** Anqi Liu, Wenfu Zhao.

**Writing – review & editing:** Anqi Liu.

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
