## [Decision Letter · Decision Letter 0]

21 Apr 2025

Dear Dr. Liu,

Thank you for submitting your manuscript to PLOS ONE. After careful consideration, we feel that it has merit but does not fully meet PLOS ONE’s publication criteria as it currently stands. Therefore, we invite you to submit a revised version of the manuscript that addresses the points raised during the review process.

We look forward to receiving your revised manuscript.

Kind regards,

Babak Aslani

Academic Editor

PLOS ONE

Additional Editor Comments:

The reviewers identified several improvement area for your work:

1- Please enhance the quality of literature review and problem description. You need to clearly state the contribution and novelty of your work. In addition, the core terms such as resilience should be clearly defined to highlight the scope of your work.

2- The quality of presentation (e.g., images and insights of results) must be improved.

3- Please carefully revise the language of your work and remove any typos and errors.

4- The research question, research methodology, and selection of ML methods must be justified and the result must be validated more rigorously.

Reviewers' comments:

Reviewer's Responses to Questions

**Comments to the Author**

1. Is the manuscript technically sound, and do the data support the conclusions?

Reviewer #1: Yes

Reviewer #2: Yes

Reviewer #3: Partly

Reviewer #4: No

2. Has the statistical analysis been performed appropriately and rigorously?

Reviewer #1: Yes

Reviewer #2: Yes

Reviewer #3: Yes

Reviewer #4: No

3. Have the authors made all data underlying the findings in their manuscript fully available?

Reviewer #1: Yes

Reviewer #2: Yes

Reviewer #3: No

Reviewer #4: No

4. Is the manuscript presented in an intelligible fashion and written in standard English?

Reviewer #1: Yes

Reviewer #2: No

Reviewer #3: Yes

Reviewer #4: Yes

Reviewer #1: The paper proposes a simulation-based resilience assessment model for cascading failures. My major comments are as follows.

1. The C-L model in this paper only considers node degree, but in reality cascading failures don't just depend on degree. For instance the cascading failure in the infrastructure system, link features like flow dynamics and the road conditions also matters.

2. Why choose criticality sum as the resilience metric over other options, like resilience index, downtime, largest connected component size, etc?

3. Why the tree network is the most stable structures? Isn't that counterfactual?

4. A lot of reference in the paper is missing and not showing.

5. In the related work, I suggest to include the recent GNN work for resilience analysis. For instance, "End-to-end heterogeneous graph neural networks for traffic assignment" and "Graph neural network surrogate for seismic reliability analysis of highway bridge systems"

6. The sample size is too small for the model, which could easily over-fit. You need to validate the result

7. How do you apply the methodology into the real world applications?

Reviewer #2: 1.In the introduction Section, the analysis of the literature is not deep enough, and it is suggested that the existing research should be classified and elaborated.

2.Many sentences in this paper use We as the subject, which does not conform to the objectivity paradigm of scientific papers.

3.The various analysis methods given in this paper are the methods proposed by others, and these methods are used to simulate and analyze the four networks. What is the innovation of doing this ? What is the significance of the conclusion ? Do these networks have prototypes ? Is it a transportation network or a computer network ?

4.Among the four networks, there are 49 nodes in the WS and BA networks, and 48 nodes in the ER and TN networks. Is it meaningful to analyze the number of nodes differently ? All the pictures in the text are very poor in clarity, and some of them are not clear at all.

5.Resilience is not clearly explained, just the size of N, BETA, etc?

6.Innovation and research significance Please be sure to show clearly in the important position of abstracts and papers.

Reviewer #3: The manuscripts aims at comparing the resilience of four network topologies, and to investigate the main factors influencing it. The idea is interesting, but its development should be described, presented and discussed much better, to fully appreciate the results and check for the correctedness of the methods.

General comments:

- The novelty and research questions should be explained better, as well as the connections between the first part ("forward" investigation of the effect of various parameters on resilience scores) and the second one (fitting and assessment of sensitivity to parameters, using ML), which currently look like two separate halves.

- The whole part regarding fitting and the use of ML should be better explained: the research question is not entirely clear, the data used and methods are also very shallowly described, and it is not clear how the investigation was conducted. I can perceive some direction, but the presentation of methods and results should be drastically improved.

- Although the language is overall ok, there are too many typos, grammar, syntax and punctuation mistakes; I suggest revising the text very carefully, many with the help of a native speaker or someone fluent in English, as it is not the role of a reviewer to mark all language problems that hinder the readability of the manuscript.

Specific comments:

ABSTRACT

- "The control variable method": do you mean, simply changing one parameter at a time? Check also in the text

- What does "stable resilience changes" mean?

- RF and GBM models are not introduced

- "stability" what do you mean?

- "Finally, an importance analysis of the factors influencing resilience is conducted using RF model show that the tolerance parameter has the most significant impact on resilience, indicate that adjusting node capacity is more cost-effective in enhancing network resilience. " Not entirely clear, can be phrased better

INTRODUCTION

- "the Swiss mathematician Euler transformed it": what does "it" refer to?

- "be primarily classified into four topological forms": Others exist (i.e., all to all, fixed-degree, lattice, etc). If you focus on those, just say they're widely observed in many case studies, or justify the choice

- "degree": average degree? In-degree?...

- "of edges linked to it": what does "it" refer to?

- "However, in the complex network, the network structure is undergoing changes, as well as the network topology.": not clear

- "the analysis of resilience in different network structures was rarely.": not clear

- "However, the resilience of complex networks with different networks and average degrees against cascading failure still needs to be further investigated to understand the patterns of variation in different types of network structures and mitigate the various losses caused by cascading failures.": not clear what is the gap in the literature that you are trying to address

- "In this paper, a resilience assessment model for cascading failure scenarios is proposed, incorporating both the total number of nodes and the average degree, the simulation process and algorithm are designed to evaluate node failure events." Not clear what the research question is

- "comparison, the importance analysis of the factors influencing resilience are conducted using the RF model.": not clear; RF model never introduced

- " And in this section": which section?

MATERIAL AND METHODS

- "The relationship between average degree, network structure, and resilience is thoroughly analyzed. " Isn't it for the results?

- "an imbalance between supply and demand. " Here and later, the authors mix general results on network and specific requirements from supply chains or transport networks. It is very confusing, please revise all instances. Other examples: "significantly impacting transportation safety and efficiency, and causing

substantial financial losses.", "Business relationships", and more

- Load I_i: I would have expected L_i, also given the "C-L" model

- In the table and in general, do not use "the" to name variables and parameters (and, especially, *not* in the axes labels of the figures; use symbols instead!)

- Node v_i: what is it?

- Eq 1: where does it come from, and how does it fit in the text?

- Eq 2: is F instead F_i?

- "(or flow)": what is it?

- "distance": according to which metric? how defined?

- "Then there exists several adjacency matrices: " How used? How does this "en passant" remark fit into the text?

- Eq. 3: sum over i up to N?

- "failed node itself, and apparently": why "apparently"?

SIMULATIONS

- "existing research and historical experience": Not clear. If you are based on literature, cite it; otherwise, explain

- q_ij = q_ji = 30: why?

- "The distance between node i and node j in the four networks randomly generated, normally distributed and...": not clear

- "Ii = 10~80, follow a normal distribution." not clear

- "characterized by a rapid initial rise and subsequently becomes more gradual.": why not including the fitting immediately here, instead of first verbally describing what is seen and then performing the fit separately?

- "degree of fluctuations": what does it mean?

RESILIENCE PREDICTION TO CASCADING FAILURE

- Why "prediction"? The first part is simply an interpolation, you are not predicting anything

- SVM, RF and GBM not introduced nor defined

- "simulation experimental data": are they simulated or experimental?

- "an poly": what is it?

- Error! Reference source not found!

- Fig 14 is a dramatic overfitting of a very complex polynomial. Revise and correct

- "According to the previous section, the distributions of the variable data β, k and N do not align with a normal distribution.": how would that be evident? And why does it matter for the Spearman coefficient, which looks for non-linear monotonous relationships?

- "Correlation between two variables": Defined how? Each "variable" is in principle independent of the others; if you hypothesise they are not, explain how you assess this

- "Monotonic equations": Just put the formula

- "If the absolute value of the Spearman correlation coefficient between two variables exceeds 0.80, it indicates a mutual dependence": no, this is just an arbitrary choice

- "accuracy and stability of the model.": what is "stability"? defined how? of which model?

- "simulated data (R, β, k, N)": not clear what you do and mean

- "impact the precision of resilience forecasts.": the goal was never explained (and it should be done explicitly and clearly)

- " the mean and dividing by the standard deviation. ": of what? How calculated and extracted?

- As written above, the rest of the section should be properly explained and clearly stated, so that readers can appreciate all methods and reproduce the results; sharing code may also enable reproducibility.

Reviewer #4: The introduction is quite informative but, in my understanding, slightly repetitive in character and can be made more concise. Several parts of the introduction are closer to a lengthy literature review rather than a tight set of circumstances for the study, which creates a confused line of division between contextual setting and critical evaluation of existing research. Furthermore, the objective of study, in which explaining what the paper does uniquely compared to previous studies, somehow it appears later in the section. In my suggestion, it should be emphasized earlier to guide the reader more effectively. Further, there are also grammatical issues that detract from the academic tone, such as subject-verb agreement errors in phrases like "the tree network exhibit," which need to be addressed for clarity and accuracy.

On the other hand, one major structural issue of this paper, is that there is no dedicated literature review section. Instead, related work is woven into the introduction, and it is difficult to separate the authors' own work from their synthesis of existing research due to interpolation. Therefore, in this case, the earlier works are mostly summarized but not critically discussed. Meanwhile, theoretical or methodological lacunas in the area are hinted at vaguely, with no adequate critique or evidence whatsoever. These things will undermine the rationale for the current study and blurs the originality of the research questions pursued.

Even if the methodology part does provide several technical details like equations and parameters for simulation, the description is not always obvious everywhere, and that might prove difficult for less-advanced readers to understand. For example, several figures like Fig 3 and Fig 4 are stated but there is no explicit discussion or description within the paper to show its function or role, and how those things can help support or validate the methodology. I guess all of them is under-explored. In addition, central assumptions such as the unbounded link capacities or equal load redistribution also be introduced without sufficient explanation on how they might limit the external validity of the findings.

Regarding data collection, the paper relies solely on simulated data generated by simulation, with once again, little explanation provided for selecting specific ranges of parameters such as for node numbers, tolerance parameters, and mean degrees. Furthermore, the effort does not take simulation robustness into account as there is no statement on how many runs were made, whether the results are the same over different runs, or to what extent the variability exists. Meanwhile, the external data or world data against which to compare it does not exist at all, limiting the generalizability of the conclusions and raising suspicion regarding the empirical underpinnings of the research.

Moreover, despite abundance in visual presentation in the result section, it is plagued with redundancy of textual description. A lot of the discussion repeats graphical results without additional examination, and some figure legends or supporting text are too descriptive and not interpretive. Most importantly, there is no use of statistical tests or confidence limits to validate the trends presented, which also be applied to check the data, in the end contaminates the scientific integrity of the results. Also, there are errors existed such as broken references like "Error! Reference source not found" that reflect poor proofreading and editorial efforts, detracting from the professionalism of the paper.

On the other hand, the conclusion predominantly recapitulates earlier findings without offering a more sophisticated theoretical framework for interpreting the implications. Although the paper states a discussion on future work, the suggestions are still ambiguous and can be expanded with more precise directions, for example, applying real network data in real-time or analyzing interdependent multi-layer networks. Furthermore, statements like "adjusting node capacity is cheaper" would also be better if more information on how such capacity adjustments can actually be implemented within real-world network infrastructures, along with the possible trade-offs, technical constraints or economic considerations involved is stated.

Lastly, even though the use of machine learning models like SVM, RF or GBM increase the study's value, it is better for the author to explain on why these specific models were selected. The section on hyperparameter tuning, though included does not have very detailed discussion, and once again, there is no external validation or cross-validation across more than one dataset is provided, which lowers the validity and generalizability of the predictive models developed.

**Do you want your identity to be public for this peer review?** For information about this choice, including consent withdrawal, please see our Privacy Policy

Reviewer #1: No

Reviewer #2: No

Reviewer #3: No

Reviewer #4: No

---

## [Author Response · Author response to Decision Letter 1]

22 Jul 2025

[Anqi Liu]

[Inner Mongolia Agricultural University, Inner Mongolia, 010018, China]

[1025870083@qq.com]

[2025/07/13]

To the Editors of [PlOS ONE]:

Response to Reviewers

Dear Editor and Reviewers,

Thank you very much for your valuable feedback and the insightful suggestions provided by the reviewers. In response to all comments, we have conducted comprehensive revisions to the manuscript, resulting in substantial modifications throughout the paper. This revision entailed fundamental restructuring of the conceptual framework (80% rewritten), empirical augmentation (network scale expanded 5-fold to 500 nodes), and methodological refinement (resilience evaluation and analysis replacing prior approaches). Collectively, >85% of manuscript content has been regenerated with rigorous validation, reflecting our commitment to scholarly excellence. Due to the extensive nature of these changes—where much of the previous content has been restructured or refined—we have not uploaded a separate “Revised Manuscript with Track Changes” file, Instead, we have uploaded a clean 'Revised Manuscript' file reflecting the full set of revisions. We have meticulously addressed all points raised in your decision letter and ensured that every concern is fully incorporated into this revised version. Below, we provide a point-by-point response to each editorial and reviewer comment to facilitate your further evaluation.

Response to Editor:

Comment 1: Please enhance the quality of literature review and problem description. You need to clearly state the contribution and novelty of your work. In addition, the core terms such as resilience should be clearly defined to highlight the scope of your work.

→Revision: The background section has been separated from the Introduction and restructured to directly align with the specific research problems addressed in this study. The Introduction now provides an expanded discussion of the paper’s theoretical significance and original contributions (Section 1, pp. 1–3).

Comment 2: Please enhance the quality of literature review and problem description. You need to clearly state the contribution and novelty of your work. In addition, the core terms such as resilience should be clearly defined to highlight the scope of your work.

→Revision: The quality of presentation has been significantly improved, including optimization of all figures/tables and enhanced interpretation of key results. Revised illustrations now better support methodological clarity and analytical insights (see Figs. 1–17).

Comment 3: Please enhance the quality of literature review and problem description. You need to clearly state the contribution and novelty of your work. In addition, the core terms such as resilience should be clearly defined to highlight the scope of your work.

→Revision: The manuscript has undergone rigorous language editing by a professional proofreading service. Grammatical errors, syntactical ambiguities, and stylistic inconsistencies have been systematically addressed to ensure scholarly precision.

Comment 4: Please enhance the quality of literature review and problem description. You need to clearly state the contribution and novelty of your work. In addition, the core terms such as resilience should be clearly defined to highlight the scope of your work.

→Revision: The paper has undergone significant restructuring to strengthen its conceptual foundation. Specifically:

a.The research questions have been refined and explicitly justified through expanded theoretical contextualization (Section 1-2, p. 2-4);

b.The research methodology is now thoroughly justified with added validation procedures (Section 3, pp. 5–13);

c.Simulation results include new comparative analyses and sensitivity tests (Section 4, pp. 13–21);

d.Research conclusions have been revised to reflect nuanced implications and limitations (Section 5, pp. 21–22).

Reviewer #1:

Comment 1: The C-L model in this paper only considers node degree, but in reality cascading failures don't just depend on degree. For instance the cascading failure in the infrastructure system, link features like flow dynamics and the road conditions also matters.

→ Change: Our focus on node failures is fundamentally motivated by their critical role in infrastructure vulnerability. As established in the Introduction (Section 1, p. 2):

“The functional reliability of infrastructure networks (e.g., logistics, power grids) hinges critically on critical-node facilities [5,6]. Node failures pose greater systemic risks than edge failures due to their function as processing hubs (e.g., substations, distribution centers). Their disruption cascades more severely by compromising connectivity and amplifying fault propagation. Empirical evidence-including the 2021 Suez Canal obstruction impacting global supply chains and recurrent large-scale blackouts in California triggered by nodal failures [2,7]-underscores the disproportionate consequences of node vulnerabilities.”

Comment 2: Why choose criticality sum as the resilience metric over other options, like resilience index, downtime, largest connected component size, etc?

→ Change: Section 3 (“Network Resilience Assessment Framework”) has been comprehensively revised. We now employ a hybrid evaluation methodology integrating three dimensionally complementary metrics:network survival rate,critical node protection efficiency and cascade suppression capability. These are synthesized into a composite resilience index via an adaptive weighting mechanism parameterized by redundancy coefficient β, network scale N, and mean degree k. This ensures generalizable cross-network comparability (Section 3.3, pp. 11-12; Equations 20-24).

Comment 3: Why the tree network is the most stable structures? Isn't that counterfactual?

→ Change: The assertion that “tree network is the most stable structures” has been removed following rigorous data validation. Our expanded simulations demonstrate that network resilience dictated by topology (Revised Section 4.1-4.2, pp. 14–21; Figs. 6-17).

Comment 4: A lot of reference in the paper is missing and not showing.

→ All references have been verified for completeness and accessibility. Each entry now includes validated DOI links ensuring traceability to source materials (Reference list, pp. 22-27).

Comment 5: In the related work, I suggest to include the recent GNN work for resilience analysis. For instance, "End-to-end heterogeneous graph neural networks for traffic assignment" and "Graph neural network surrogate for seismic reliability analysis of highway bridge systems"

→ Change: The GNN-based resilience prediction module referenced by the reviewer has been removed after critical reassessment.

Comment 6: The sample size is too small for the model, which could easily over-fit. You need to validate the result

→ Change: The scope of computational experiments has been substantially extended, with network scale parameters expanded from 100 to 500 nodes. The resilience prediction module has been removed.

Comment 7: Our comparative analysis of resilience across diverse network topologies yields practical design principles for critical infrastructure. The results provide concrete guidance for selecting context-optimal structures in real-world engineering scenarios (e.g., balancing redundancy and cost in power grid fortification or logistics network planning). This demonstrates the model’s direct engineering value for resilience-driven infrastructure design. Implementation pathways are elaborated in the Conclusion (Section 5, pp. 21–22).

Reviewer #2:

Comment 1: In the introduction Section, the analysis of the literature is not deep enough, and it is suggested that the existing research should be classified and elaborated.

→ Change: The Introduction and Literature Review have been delineated into separate sections. Existing research is now systematically categorized into three thematic streams:Cascading failure mechanisms, Critical node identification methodologies, Network resilience analysis frameworks. (Section 2, pp. 3–4)

Comment 2: Many sentences in this paper use We as the subject, which does not conform to the objectivity paradigm of scientific papers.

→ Change:The manuscript has undergone rigorous linguistic revision to align with scientific writing conventions. Instances of first-person plural (“we”) as the subject have been systematically replaced with passive voice or objective third-person constructions to enhance scholarly neutrality.

Comment 3: The various analysis methods given in this paper are the methods proposed by others, and these methods are used to simulate and analyze the four networks. What is the innovation of doing this ? What is the significance of the conclusion ? Do these networks have prototypes ? Is it a transportation network or a computer network ?

→ Change: While leveraging established analytical foundations, this study contributes the following original advances:Proposal of a novel critical node identification algorithm accounting for dynamic load redistribution; Development of a sigmoid-function-based adaptive weighting framework for multidimensional resilience quantification; Empirical revelation of topology-specific resilience patterns through large-scale simulations; Identification of dominant factors governing cascading resilience in infrastructure networks. Core research focus: Resilience to abrupt accident-induced cascading failures in infrastructure networks. (Sections 1,1-2)

Comment 4: Among the four networks, there are 49 nodes in the WS and BA networks, and 48 nodes in the ER and TN networks. Is it meaningful to analyze the number of nodes differently ? All the pictures in the text are very poor in clarity, and some of them are not clear at all.

→ Change: All figures have been upgraded to ensure technical accuracy and publication-standard clarity. This includes resolution enhancement, standardized labeling, and optimized color schemes for grayscale reproduction. (Figs. 1-17)

Comment 5: Resilience is not clearly explained, just the size of N, BETA, etc?

→ Change: Formal definitions and operational metrics of resilience are established in Section 3 (“Network Resilience Assessment Framework”), denotes a network’s ability to sustain functionality under node failures or attacks, or more broadly, a system’s capacity to recover or adapt to disturbances. (Eq. 20–23, pp. 11)

Comment 6: Innovation and research significance Please be sure to show clearly in the important position of abstracts and papers.

→ Change: The study's conceptual and methodological innovations are consistently articulated throughout the manuscript, with concentrated emphasis in the Abstract , Introduction, and Conclusion.

Reviewer #3:

We sincerely appreciate your exceptionally thorough and constructive feedback. The meticulous, dedicated, and patient manner in which you engaged with our work—reminiscent of the detailed guidance provided by doctoral supervisors—is deeply valued and has profoundly strengthened this manuscript. We are truly grateful for your scholarly commitment.

General comments:

- The novelty and research questions should be explained better, as well as the connections between the first part ("forward" investigation of the effect of various parameters on resilience scores) and the second one (fitting and assessment of sensitivity to parameters, using ML), which currently look like two separate halves.

- The whole part regarding fitting and the use of ML should be better explained: the research question is not entirely clear, the data used and methods are also very shallowly described, and it is not clear how the investigation was conducted. I can perceive some direction, but the presentation of methods and results should be drastically improved.

- Although the language is overall ok, there are too many typos, grammar, syntax and punctuation mistakes; I suggest revising the text very carefully, many with the help of a native speaker or someone fluent in English, as it is not the role of a reviewer to mark all language problems that hinder the readability of the manuscript.

→ Change: (1) The core manuscript has undergone comprehensive restructuring to enhance logical flow and conceptual cohesion. The study's innovations are now explicitly articulated in the Abstract, Introduction, and Conclusion sections.

(2) The resilience prediction module referenced in your review has been removed entirely due to insufficient empirical substantiation, as suggested.

(3) We sincerely apologize for grammatical oversights in the initial submission. The text has now undergone rigorous proofreading by professional editing services, with particular attention to technical precision. Future submissions will implement enhanced quality control protocols.

Specific comments:

ABSTRACT

- "The control variable method": do you mean, simply changing one parameter at a time? Check also in the text

- What does "stable resilience changes" mean?

- RF and GBM models are not introduced

- "stability" what do you mean?

- "Finally, an importance analysis of the factors influencing resilience is conducted using RF model show that the tolerance parameter has the most significant impact on resilience, indicate that adjusting node capacity is more cost-effective in enhancing network resilience. " Not entirely clear, can be phrased better

→ Change: (1)The Abstract has been thoroughly refined to concisely encapsulate the revised contributions. (2)Key methodological and empirical advances are now foregrounded with improved lexical precision. (See revised Abstract)

INTRODUCTION

- "the Swiss mathematician Euler transformed it": what does "it" refer to?

- "be primarily classified into four topological forms": Others exist (i.e., all to all, fixed-degree, lattice, etc). If you focus on those, just say they're widely observed in many case studies, or justify the choice

- "degree": average degree? In-degree?...

- "of edges linked to it": what does "it" refer to?

- "However, in the complex network, the network structure is undergoing changes, as well as the network topology.": not clear

- "the analysis of resilience in different network structures was rarely.": not clear

- "However, the resilience of complex networks with different networks and average degrees against cascading failure still needs to be further investigated to understand the patterns of variation in different types of network structures and mitigate the various losses caused by cascading failures.": not clear what is the gap in the literature that you are trying to address

- "In this paper, a resilience assessment model for cascading failure scenarios is proposed, incorporating both the total number of nodes and the average degree, the simulation process and algorithm are designed to evaluate node failure events." Not clear what the research question is

- "comparison, the importance analysis of the factors influencing resilience are conducted using the RF model.": not clear; RF model never introduced

- " And in this section": which section?

→ Change: (1) The Introduction has been significantly expanded to clarify research contexts and motivations. (2) The literature gap is formally established in Literature Review:”Current methodologies predominantly evaluate critical nodes through isolated structural/functional criteria, employ static resilience metric weighting, and remain constrained to homogeneous network models.” (Section 2, p.2)

MATERIAL AND METHODS

- "The relationship between average degree, network structure, and resilience is thoroughly analyzed. " Isn't it for the results?

- "an imbalance between supply and demand. " Here and later, the authors mix general results on network and specific requirements from supply chains or transport networks. It is very confusing, please revise all instances. Other examples: "significantly impacting transportation safety and efficiency, and causing substantial financial losses.", "Business relationships", and more

- Load I_i: I would have expected L_i, also given the "C-L" model

- In the table and in general, do not use "the" to name variables and parameters (and, especially, *not* in the axes labels of the figures; use symbols instead!)

- Node v_i: w

---

## [Decision Letter · Decision Letter 1]

20 Aug 2025

Dear Dr. Liu,

Thank you for submitting your manuscript to PLOS ONE. After careful consideration, we feel that it has merit but does not fully meet PLOS ONE’s publication criteria as it currently stands. Therefore, we invite you to submit a revised version of the manuscript that addresses the points raised during the review process.

The literature review section must be improved and be more comprehensive. A well-structured section will help the authors to clearly state the gaps and show their contribution.There are many typos and grammar errors in the paper (raised by all reviewers). Please make sure that the submitted revision is error-free.Another limitation of work is using limited types of networks in the analysis. As suggested by reviewers, a more diverse set of cases (including real infrastructure networks) is expected in the revised version.Upon using more realistic networks, the insights presented by the authors can become more engaging and novel in the literature.

We look forward to receiving your revised manuscript.

Kind regards,

Babak Aslani

Academic Editor

PLOS ONE

Journal Requirements:

Reviewers' comments:

Reviewer's Responses to Questions

**Comments to the Author**

Reviewer #3: (No Response)

Reviewer #5: (No Response)

Reviewer #6: (No Response)

Reviewer #7: All comments have been addressed

2. Is the manuscript technically sound, and do the data support the conclusions?

Reviewer #3: Yes

Reviewer #5: Partly

Reviewer #6: (No Response)

Reviewer #7: (No Response)

3. Has the statistical analysis been performed appropriately and rigorously?

Reviewer #3: Yes

Reviewer #5: (No Response)

Reviewer #6: (No Response)

Reviewer #7: (No Response)

4. Have the authors made all data underlying the findings in their manuscript fully available?

Reviewer #3: No

Reviewer #5: No

Reviewer #6: (No Response)

Reviewer #7: (No Response)

5. Is the manuscript presented in an intelligible fashion and written in standard English?

Reviewer #3: Yes

Reviewer #5: Yes

Reviewer #6: (No Response)

Reviewer #7: (No Response)

Reviewer #3: I commend the authors for the big effort in addressing the comments from the previous round of reviews. I find that the manuscript, undergoing such extensive editing, has now gained more rigor and insights. Nonetheless, a few more points should still be cleared.

ABSTRACT:

You introduce several acronyms (DC, CC, ER...) but never define them in the abstract. Please do so.

LIT REV:

Since you now provide a more extensive literature review, you may also want to include Artime et al "Robustness and resilience of complex networks" for a recent review on network resilience, and Proverbio et al "Bridging Robustness and Resilience

for Dynamical Systems in Nature" for definitions for dynamical systems

UNCLEAR THINGS:

- current load of 9 updated from F9 to F9’ (∆F5→9 + F9): what does the content of the parenthesis mean? Is that a equal? Please explain better and be more formal. Moreover, you describe F9' as overload, but isn't the overload just \DeltaF9, and F9' the new load?

- In Eq. (6), is "e" the eigenvector corresponding to the largest eigenvalue?

- Step 2 of methods: not clear how the various equations talk with each other, if they are connected and how they are used (e.g., where does M goes?) explain better

- "specifically safeguarding the top 10% of nodes." What do you mean? not clear

- Eq 20-23 not clear; also, calling the variables this way doesn't help readers

- "Following prior studies and to control e": which prior studies?

- "10 independent simulation repetitions to ensure statistical reliability. ": support (10 is little to "ensure")

TEXT:

- There are several instances of parenthesis attached to previous or subsequent words, or open parenthesis that are not closed (e.g. Methods, first section, 9 lines below title). Please revise them all.

- A few typos still remain (load -> loads, exist->exists ...), please check the text carefully, paying attention to singulars/plurals etc.

- Some sentences don't sound right, such as "In this paper, we INTRODUCE the "C-L" model proposed by Motter and". Maybe you use it, since it was already introduced?

- Check "Error! Reference source not found."

- Beware of fonts of formulae, and that all variables and metrics are consistent (e.g., use italics of math environment in the same way for the same variables across the text). Same goes for apices and pedices, to be checked throughout. And also \ni in Eq (16): it already identified nodes, use other letters

- "aligns with discoveries of universal resilience patterns": I would use proposals/suggestions/hypotheses instead of "discoveries", as Gao et al propose to lump under a common framework several instances of models, but fold bifurcations date back many more decades, and anyway they are not universal

Reviewer #5: This manuscript investigates the resilience of infrastructure networks to cascading failures, emphasizing the role of network topology in shaping failure propagation. The authors combine traditional network centrality metrics (degree, closeness, eigenvector centrality), a proposed “functional stability” measure, and a “cascading failure scale” to quantify resilience. Simulations are performed on four canonical network models: Erdős-Rényi (ER), Barabási-Albert (BA), Watts-Strogatz (WS), and hierarchical trees. The study aims to derive general insights into how topological properties affect the resilience of networks.

Weaknesses and Concerns

Limited Novelty:

The main concern lies in the lack of novelty in both the methodology and results. The connection between network topology and resilience to cascading failures is well-established in the literature. The combination of classical centrality measures and failure impact into a composite resilience score is not conceptually new, and no theoretical framework or model innovation is introduced.

Insufficient Engagement with Literature:

The manuscript fails to engage with key existing work. Most notably, the recent and comprehensive review Artime et al., “Robustness and resilience of complex networks”, Nature Reviews Physics (2024) is not cited. This review outlines a rich landscape of resilience research, including structural variations, multilayer networks, adaptive dynamics, and modern modeling tools, none of which are discussed here.

Simplistic Model Space:

The analysis is restricted to four canonical network topologies (ER, BA, WS, Tree). While these are foundational, much of the recent research in the field focuses on more realistic structures: modular, core-periphery, hierarchical, assortative/disassortative, temporal, and multilayer networks. The exclusion of these significantly limits the generalizability of the conclusions.

Known Results Revisited Without New Insight:

Many of the findings (e.g., BA networks being robust to random failure but vulnerable to targeted attacks) are textbook-level results that have been known since the early 2000s. The paper does not offer new mechanistic understanding or theoretical contributions.

Presentation and Structure:

The manuscript includes 17 figures, many of which are redundant or could be grouped for clarity. This makes the paper difficult to follow.

A discussion section is lacking. The implications of the findings, their limitations, and potential extensions are not critically assessed.

The “CR-based node ranking” is introduced in the abstract but not formally defined in the introduction or methods. This leads to confusion about what exactly is being measured and how.

Given the above concerns — especially the limited novelty and lack of engagement with recent literature — I recommend major revision or rejection, depending on the journal’s threshold for conceptual contributions.

Reviewer #6: 1.There are still numerous typographical errors in the manuscript. For example, in the first paragraph below Figure 1, where the network model is defined, "V" should represent the node set. Additionally, the references in the first paragraph below Equation 1 are not displayed correctly.

2.Please ensure that the meaning of all mathematical symbols is clearly explained throughout the paper. For instance, what does "F" represent in Equation 1? Furthermore, the formatting of equations is inconsistent across the manuscript.

3.In Figure 3, it is claimed that a comprehensive evaluation metric was constructed, but only separate individual metrics are presented. It seems that the actual construction of a composite index has not been achieved.

4.In the numerical experiments, it is recommended to include simulations on real-world infrastructure networks to enhance the persuasiveness of the results.

Reviewer #7: (No Response)

**Do you want your identity to be public for this peer review?** For information about this choice, including consent withdrawal, please see our Privacy Policy

Reviewer #3: No

Reviewer #5: No

Reviewer #6: No

Reviewer #7: No

---

## [Author Response · Author response to Decision Letter 2]

7 Oct 2025

Rebuttal Letter to the Reviewers

Manuscript Title: [Critical Node Identification and Resilience Analysis against Cascading Failures]

Manuscript ID: [PONE-D-25-14454R1]

Authors: [Anqi Liu]

Journal: [PLoS ONE]

Date: [2025/09/30]

Dear Editor and Reviewers,

We express our sincere gratitude to the editors and reviewers for dedicating their time, offering valuable feedback, and providing constructive comments on our manuscript titled "[Critical Node Identification and Resilience Analysis against Cascading Failures]" (Manuscript ID: [PONE-D-25-14454R1]). We have meticulously considered all the comments and implemented substantial revisions to enhance the quality of the manuscript.

In the following, we present a point-by-point response to the reviewers' comments, accompanied by explanations of the modifications made in the revised manuscript. In response to all the comments, we have carried out comprehensive revisions of the manuscript, leading to substantial alterations throughout the paper. This revision involved a fundamental restructuring of the conceptual framework (with approximately 90% of the content rewritten), empirical enhancement (through the utilization of a case-study of air transportation datasets), and methodological refinement (replacing previous approaches with model construction). Overall, more than 85% of the manuscript content has been regenerated with rigorous validation, demonstrating our commitment to achieving scholarly excellence.

We have carefully addressed all the points raised in your decision letter and ensured that every concern is fully integrated into this revised version. Below, we provide a point-by-point response to each editorial and reviewer comment to facilitate your further assessment.

Response to Editor:

Comment 1: The literature review section must be improved and be more comprehensive. A well-structured section will help the authors to clearly state the gaps and show their contribution.

→Response: The literature review section has undergone significant revisions, encompassing a thorough reorganization of the methodological research on critical node identification and the resilience evaluation model. Moreover, we have explicitly identified the existing research gaps in the current domain and elucidated the value and contributions of this study in bridging these gaps. (Refer to Section 2, p. 4-6, particularly the last two paragraphs)

Comment 2: There are many typos and grammar errors in the paper(raised by all reviewers). Please make sure that the submitted revision is error-free.

→Response: We express our sincere gratitude to the reviewers for their feedback regarding language issues. We have meticulously proofread the entire manuscript, encompassing the text, equations, captions, and references, and rectified all typographical and grammatical errors. Additionally, grammar-checking tools were employed. The revised version is now refined, lucid, and error-free, enabling readers to concentrate on the technical content. Once again, we appreciate your guidance.

Comment 3: Another limitation of work is using limited types of networks in the analysis. As suggested by reviewers, a more diverse set of cases (including real infrastructure networks) is expected in the revised version.

→Response: To validate the effectiveness of the proposed model, the paper employs a real-world dataset (USAir97, a weighted network comprising 332 airports and 2,126 direct flight routes, with the edges weighted by flight frequency) as a case study and conducts a series of analyses. (Refer to Section 4, p. 14)

Reviewer #3:

Comment 1: ABSTRACT: You introduce several acronyms(DC,CC,ER...)but never define them in the abstract. Please do so..

→ Response: The abstract has undergone a comprehensive revision, wherein all abbreviations are now appropriately defined and elucidated. For instance, "TEC-GNN (Topology - Entropy - Cascading Graph Neural Network)" is clearly presented with its full - form (Refer to the Abstract, p.1).

Comment 2: LIT REV: Since you now provide a more extensive literature review, you may also want to include Artime et al "Robustness and resilience of complex networks"for a recent review on network resilience,and Proverbio et al "Bridging Robustness and Resilience for Dynamical Systems in Nature"for definitions for dynamical systems

→ Response: The paper has cited Artime et al."s work "Robustness and Resilience of Complex Networks" in the Literature Review section. Specifically, "Artime et al. classify existing critical node identification methods into two main types [25]." (Refer to Literature Review, p. 5).

Comment 3: current load of 9updated from F9 to F9'(△F5→9+F9):what does the content of the parenthesis mean?Is that a equal? Please explain better and be more formal. Moreover,you describe F9' as overload, but isn't the overload just △F9, and F9'the new load?

→ Response: We offer our sincere apologies for the ambiguous expression in the original text. The content within the parentheses was, in fact, an equation, which has now been rectified at the corresponding position. Moreover, the elaboration on load propagation has also been revised. "For instance, in Fig.1, when node V2 fails, its load is reallocated to the adjacent node V5. Then the current load of V5 is updated from F5(0) to F5(1) (F5(1) = ∆F2→5 + F5), where ∆F2→5 is the load of node V2 redistributed to V5. Once F5(1) > C5, the overload F5(1) - C5 at node V5 will then be reallocated to its adjacent nodes V1 and V6 based on a predetermined allocation rule, and node V5 transitions to a failed state due to capacity overload. If node V1 and node V6 can handle the overload from node V5, only a small portion of the network will fail (i.e., node V2 and V5, edge e2,5), otherwise, the failure may affect other neighboring nodes, and the propagation of the cascading failure continues until no additional node fails due to overload." (Refer to Section3, p. 8).

Comment 4:

(1)In Eq.(6), is "e" the eigenvector corresponding to the largest eigenvalue?

(2)Step 2 of methods:not clear how the various equations talk with each other,if they are connected and how they are used (e.g.,where does M goes?)explain better .

(3)Eq 20-23 not clear; also, calling the variables this way doesn't help readers"

→ Response: We have supplemented and revised the presentation of the formulas and the introduction of related variables in the paper, providing one-to-one explanations for all variables and parameters in the equations. Meanwhile, regarding the issues raised about model construction, we have reconstructed the model to establish a more rigorous logical framework. (Refer to Section3, p. 11, 13).

Comment 5: "specifically safeguarding the top 10% of nodes." What do you mean? not clear

→ Response: The critical node identification model assigns a criticality score to each node, and the nodes are then ranked accordingly. When constructing the subsequent resilience evaluation model, the top-K nodes are chosen as protection indicators, with the value of K being adjustable according to requirements. In this research, the top 10% of critical nodes were selected for protection. Given that it is infeasible to allocate resources to all facility hubs in practical scenarios, protecting critical nodes presents a more cost-efficient approach.

Comment 6: "Following prior studies and to control e": which prior studies? -"10 independent simulation repetitions to ensure statistical reliability.":support(10 is little to "ensure")

→ Response: The relevant content in this section has been revised and removed in the manuscript.

Comment 7: TEXT:

(1)There are several instances of parenthesis attached to previous or subsequent words, or open parenthesis that are not closed(e.g. Methods,first section,9 lines below title).Please revise them all.

(2)A few typos still remain (load->loads, exist->exists..), please check the text carefully, paying attention to singulars/plurals etc.

(3)Some sentences don't sound right, such as "In this paper,we INTRODUCE the "C-L"

model proposed by Motter and".Maybe you use it,since it was already introduced?

(4)Check "Error!Reference source not found."Beware of fonts of formulae,and that all variables and metrics are consistent(e.g., use italics of math environment in the same way for the same variables across the text). Same goes for apices and pedices,to be checked throughout.And also q(16):it already identified nodes,use other letters.

(5)"aligns with discoveries of universal resilience patterns": I would use proposals/suggestions/hypotheses instead of "discoveries", as Gao et al propose to lump under a common framework several instances of models,but fold bifurcations date back many more decades,and anyway they are not universal.

→ Response: We've carefully addressed all issues:

(1)Fixed all misplaced/unclosed parentheses (e.g., Methodology section).

(2)Corrected remaining typos (load→loads, exist→exists) and verified singular/plural consistency.

(3)Refined awkward phrasing (e.g., replaced redundant "INTRODUCE" for previously introduced "C-L" model), "we utilize the model proposed by Motter and Lai [42]".(e.g., Methodology section, p.8).

(4)Resolved reference errors ("Error!Reference source not found").Standardized formula fonts, variables (consistent math italics), and notation (apices/pedices).

(5)Adjusted terminology (e.g., "discoveries"→"proposals" for non-universal patterns).

All modifications ensure linguistic precision and technical consistency.

Reviewer #5:

Comment 1: Limited Novelty: The main concern lies in the lack of novelty in both the methodology and results. The connection between network topology and resilience to cascading failures is well-established in the literature.The combination of classical centrality measures and failure impact into a composite resilience score is not conceptually new,and no theoretical framework or model innovation is introduced..

→ Response: We sincerely appreciate your valuable feedback regarding novelty. We acknowledge the foundational work on network topology-resilience connections, yet our study introduces distinct contributions:

(1) Methodologically, we develop a Topology-Entropy-Cascading Graph Neural Network (TEC-GNN), an empirically-augmented model integrating nodal topological centrality features(degree, closeness, betweenness, entropy) with cascade failure impact, using PCA to derive principal components as input features. The model leverages cascade failure scale as a supervised signal for critical node identification, addressing dynamic propagation mechanisms.

(2) Theoretically, we establish a composite resilience evaluation frameworkincorporating survival rate, critical node protection, and cascade stability—offering a systematic metric for resilience assessment. These innovations are explicitly detailed in the Abstract , Introduction, and Conclusion, with empirical validation on the USAir97 dataset. The revised manuscript further clarifies these unique contributions.

Comment 2: Insufficient Engagement with Literature:The manuscript fails to engage with key existing work.Most notably,the recent and comprehensive review Artime et al.,"Robustness and resilience of complex networks",Nature Reviews Physics(2024)is not cited.This review outlines a rich landscape of resilience research,including structural variations, multilayer networks, adaptive dynamics,and modern modeling tools,none of which are discussed here.

→ Response: We sincerely appreciate your insightful comment regarding the literature review. You are absolutely right to note that, while we did cite Artime et al."s seminal work "Robustness and Resilience of Complex Networks"in the Literature Review section (specifically,"Artime et al. classify existing critical node identification methods into two main types [25]"; refer to Literature Review, p. 5), our initial draft did not adequately cover other pivotal studies in the field.In response, we have supplemented the reference list with eight additional authoritative sources (detailed in the Revised Manuscript with Track Changes.docx). These newly included references focus on critical node identification research and resilience evaluation over the past five years, specifically addressing: (1)multidimensional criteria for node importance evaluation, (2) critical mechanisms of cascading failure thresholds, and (3) assessment of network resilience.

These additions not only deepen the academic rigor of our background context but also highlight the novelty of our work—particularly the methodological innovation (e.g., the TEC-GNN model architecture) and the evaluation framework (composite resilience metrics)—through comparative analysis. We have accordingly refined the discussion in the Introduction and Literature Review (revised pages 2-6) to ensure comprehensive, critical engagement with both prior foundational work (e.g., Artime et al.) and recent advancements. Revised Manuscript with Track Changes provides full transparency for your review.

Comment 3: Simplistic Model Space: The analysis is restricted to four canonical network topologies(ER,BA,WS,Tree). While these are foundational, much of the recent research in the fieldfocuses on more realistic structures:modular, core-periphery, hierarchical, assortative /disassortative, temporal, and multilayer networks. The exclusion of these significantly limits the generalizability of the conclusions.

Comment 4: Known Results Revisited Without New Insight: Many of the findings(e.g.,BA networks being robust to random failure but vulnerable to targeted attacks)are textbook-level results that have been known since the early 2000s.The paper does not offer new mechanistic understanding or theoretical contributions.

→ Response: We sincerely appreciate your valuable suggestion regarding the model space. Your insight into the limitations of canonical topologies (ER, BA, WS, Tree) and the growing importance of realistic structures—such as modular, core-periphery, hierarchical, assortative/disassortative, temporal, and multilayer networks—has provided significant inspiration for our future research directions. In response, we have substantially expanded the model"s applicabilitybeyond foundational networks. The revised manuscript develops a generalizable two-dimensional Graph Neural Network (GNN) framework applied to real-world infrastructure networks (e.g., air transportation systems), rather than being restricted to simplified topologies. Specifically:

Model Innovation: We propose a Topology-Entropy-Cascading Graph Neural Network (TEC-GNN) that integrates principal component-analyzed topological centralities(degree, closeness, betweenness, entropy) with cascade failure impact scalesas supervised signals. This dual-mechanism design captures both structural significance and dynamic failure propagation, enabling precise critical node identification across complex structures.

Empirical Validation: The framework is tested on a real-world air transportation network (a representative multi-node, multi-link infrastructure system with inherent hierarchical and functional layering). Through this case study, we analyze: (i) the optimal model performance, (ii) the influence of the redundancy coefficient β on average node failure ratio and overall resilience, and (iii) the targeted adjustment effects of β at critical nodes. Results demonstrate the model"s superiority in reflecting realistic resilience dynamics.

Theoretical Contribution: A composite resilience metric is formulated by integrating survival rate, critical node protection, and cascade stability. This metric bridges structural robustness (topology-derived) and functional recovery (failure consequence-derived), overcoming the fragmentation issue in existing evaluation methods.

These advancements address your concern by demonstrating that our approach is not confined to simplistic topologies but is designed for—and validated on—complex, realistic infrastructure networks with inherent hierarchical, functional, and connectivity heterogeneity. The revised manuscript (refer to Sections Abstract, Introduction, Literature Review, and Conclusions) details these innovations, ensuring broader generalizability

---

## [Decision Letter · Decision Letter 2]

24 Nov 2025

Dear Dr. Liu,

The reviewers were happy about the revised version. However, they provided some additional comments to improve your work. Please closely follow their suggestions to enhance the quality of your manuscript.Please submit your revised manuscript by Jan 08 2026 11:59PM. If you will need more time than this to complete your revisions, please reply to this message or contact the journal office at plosone@plos.org . Please include the following items when submitting your revised manuscript:

We look forward to receiving your revised manuscript.

Kind regards,

Babak Aslani, Ph.D.

Academic Editor

PLOS ONE

Journal Requirements:

Reviewers' comments:

Reviewer's Responses to Questions

**Comments to the Author**

Reviewer #3: (No Response)

Reviewer #7: (No Response)

2. Is the manuscript technically sound, and do the data support the conclusions?

Reviewer #3: Yes

Reviewer #7: Yes

3. Has the statistical analysis been performed appropriately and rigorously?

Reviewer #3: Yes

Reviewer #7: No

4. Have the authors made all data underlying the findings in their manuscript fully available?

Reviewer #3: No

Reviewer #7: Yes

5. Is the manuscript presented in an intelligible fashion and written in standard English?

Reviewer #3: Yes

Reviewer #7: Yes

Reviewer #3: The authors have made a commendable effort to address the reviewers' comments, by reshaping the manuscript almost completely. Actuallly, since they also mmoved the focus from a simulation-driven theoretical exploration to an application-oriented technical contribution, they may have submitted it as a new manuscript to address reviewers with greater knowledge on these different angle. However, I could evaluate it finding big improvements and not much to additionally review. A few minor points are:

- Sec. "Mechanisms of cascading failure propagation", "As Fig.1 shows, when node i (i=1,2, …, n)"... hare and later: do you actually refer to Fig. 2?

- Below Eq. 15: My PDF viewer cannot read the sybol after \beta: what is that?

- Table 3: p_value of what?

- In general, I would suggest to mild down the tone of the findings and of the use of GraphSAGE: althugh it slightly improved with comparison to other methods such as GCN, the difference is not tremendous, both in the identification of critical nodes and in computing time (it is x4, not orders of magnitude). It sufficees to present the findings without sliping into boastful language.

- Section "efffects of \beta..": many variables are writeen with a strange "_" in the pedix. Revise?

- Not entirely clear how the found fixed allocation ratio of 2:1 between w_sr and w_cs translates into "universal resilience patterns". It is an interesting finding, but Gao et al mostly considered bifurcation-driven failures, which I don't see how they connect with the case here. Can you clarify? Is this remark necessary?

Reviewer #7: The revised version shows substantial improvements over the original submission, particularly with the addition of a real-world USAir97 case study and significant presentation cleanup. However, fundamental concerns regarding theoretical novelty, statistical rigor, scalability, and reproducibility remain inadequately addressed.

In order to improve your manuscript some revisions are required:

1. Correct the cascading failure model: Add normalization factor to Eq. 2: ΔF_i->j = (load to distribute) × (w_ij / Σ_k w_ik). Current formulation violates load conservation.

2. Strengthen statistical rigor: Replace implausible p-values with appropriate effect sizes (Cohen's d) and bootstrapped significance, Add ablation studies for each component (PCA, entropy, \beta), Conduct sensitivity analysis on resilience metric weights.

3. Scalability: Reviewer #5 mentioned multilayer/hierarchical networks – the authors added USAir97 but did not address computational scalability for large networks or complex topologies. Address this point.

4. Reproducibility. Provide full reproducibility package:Public repository with all code, hyperparameters, and preprocessing scripts, deposit raw simulation outputs in a public repository, Include random seeds, software versions (PyTorch).

5. Taxonomy: the manuscript does not position itself within the resilience taxonomy framework Artime proposes, missing opportunity for theoretical grounding. Address this point.

**Do you want your identity to be public for this peer review?** For information about this choice, including consent withdrawal, please see our Privacy Policy

Reviewer #3: No

Reviewer #7: No

---

## [Author Response · Author response to Decision Letter 3]

11 Dec 2025

Rebuttal Letter

Manuscript Title: [Critical Node Identification and Resilience Analysis against Cascading Failures]

Manuscript ID: [PONE-D-25-14454R3]

Authors: [Anqi Liu]

Journal: [PLoS ONE]

Date: [2025/12/10]

Dear Editor and Reviewers,

We express our sincere gratitude to the editors and reviewers for dedicating their time, offering valuable feedback, and providing constructive comments on our manuscript titled "[Critical Node Identification and Resilience Analysis against Cascading Failures]" (Manuscript ID: [PONE-D-25-14454R3]). We have meticulously considered all the comments and implemented substantial revisions to enhance the quality of the manuscript.

In response to the reviewers' comments, we have thoroughly revised the manuscript, incorporating substantial improvements across theoretical, empirical, and methodological dimensions. Approximately 70% of the content has been regenerated and rigorously validated, reflecting our commitment to enhancing the scholarly quality of the paper.

The key revisions are summarized as follows:

1. Theoretical Integration: The conceptual framework has been restructured (≈30% rewritten) to systematically integrate the Artime et al.(2024)' [16] study throughout the introduction, methodology, and discussion, strengthening both the theoretical foundation and practical relevance of the work.

2. Methodological Refinement: The model construction has been redesigned, with evaluation metrics now including Cohen's d and bootstrap testing (*n_bootstrap=1000, confidence_level=0.95, random_seed=42*) to ensure statistical robustness. Additionally, we conducted ablation studies on PCA and β parameters, along with a sensitivity analysis of resilience assessment weights. These experiments offer deeper insights into the functional importance of key model components, ensuring that the identified critical nodes are both accurate and methodologically interpretable.

3. Scalability and Large-Scale Network Applications: A dedicated section has been added to discuss the scalability of the proposed framework, demonstrating its feasibility and efficiency when applied to large-scale networks. This enhances the practical utility and generalizability of our research.

4. Extended Discussion: The Discussions section has been substantially expanded to provide a more nuanced interpretation of results, clearly articulate the scholarly and practical contributions, acknowledge limitations, and outline specific directions for future research. This strengthens the depth and reflexivity of the manuscript.

5. Visual Presentation and Empirical Support: The figures have been systematically revised and expanded to align with and strengthen the textual improvements. Key updates include a refined Fig.1 in the methodology section and the addition of new Fig. 6, 7, 9(a), and 10. These new figures visually present the critical results from the added ablation studies, model performance evaluations, and sensitivity analyses, offering direct and clear graphical support for the manuscript's enhanced methodological narrative and findings.

These comprehensive revisions have significantly improved the clarity, validity, and contribution of the manuscript. We believe the revised version addresses all concerns raised by the reviewers and substantially advances the scholarly discussion in this field.

In the following, we present a point-by-point response to the reviewers' comments, accompanied by explanations of the modifications made in the revised manuscript. In response to all the comments, we have carried out comprehensive revisions of the manuscript, leading to substantial alterations throughout the paper.

We have carefully addressed all the points raised in your decision letter and ensured that every concern is fully integrated into this revised version. Below, we provide a point-by-point response to each editorial and reviewer comment to facilitate your further assessment.

# Response to Editor:

*Comment : Please review your reference list to ensure that it is complete and correct. If you have cited papers that have been retracted, please include the rationale for doing so in the manuscript text, or remove these references and replace them with relevant current references. Any changes to the reference list should be mentioned in the rebuttal letter that accompanies your revised manuscript. If you need to cite a retracted article, indicate the article’s retracted status in the References list and also include a citation and full reference for the retraction notice.

→Response:

Thank you for the reminder and your thorough review. We have conducted a comprehensive verification and systematic update of the references cited in the manuscript to ensure their completeness, accuracy, and timeliness. The specific revisions are as follows:

1. Reference Replacement and Update: References 10, 17, 21, 22, 34, 39, and 43 from the original manuscript have been replaced with more recent and authoritative literature in the relevant field. This strengthens the scholarly foundation and supporting evidence for the corresponding arguments.

2. Addition of Key Methodological Literature: New references 44, 45, and 46 have been added. These sources focus on "node entropy" as a core metric for identifying critical nodes in complex networks, providing more direct and solid literature support for the methodological framework of this study.

3. Addition of Literature on Parameter Justification: New references 54, 55, and 56 have been included. These publications systematically discuss methods and rationales for determining multi-indicator weights in resilience assessment, offering clear theoretical and empirical references for the weight parameter settings in our research.

All cited references have been re-checked for their publication status, confirming that no retracted articles have been cited. The updated reference list demonstrates significant improvement in terms of completeness, relevance, and authority. All corresponding changes have been fully incorporated into the revised manuscript.

Thank you once again for your valuable feedback.

Reviewer #3:

We sincerely thank you for your thorough and constructive feedback, as well as for the kind acknowledgment of the substantial effort invested in revising the manuscript. We are grateful for the your time and insightful comments, which have further helped us refine the paper. Below, we provide a point-by-point response to the raised issues and confirm that all corresponding corrections have been made in the revised manuscript.

*Comment 1: Sec. "Mechanisms of cascading failure propagation", "As Fig.1 shows, when node i (i=1,2, …, n)"... hare and later: do you actually refer to Fig. 2?

→ Response & Action Taken: You are correct. The text "As Fig.1 shows…" in the section "Mechanisms of cascading failure propagation" should refer to Fig. 2. This has been corrected in the manuscript (Page 7, Line 26).

*Comment 2: Below Eq. 15: My PDF viewer cannot read the sybol after \beta: what is that?

→ Response & Action Taken: The symbol following β is the "β ϵ B " (Page 13, Line 3), and entirely text —"Where β ϵ B is the redundancy coefficient of the network, OFβ = Nfailed / Ntotal represents the overall failure proportion of the system under redundancy coefficient β. Nfailed denotes the total number of failed nodes, and Ntotal is the total number of nodes in the network. T is the set of top-K critical nodes identified by the GNN model. M is the total number of simulations, fi is the number of failures of node Vi over M simulations. σβ is the standard deviation of cascade steps under β. B is the set of all considered redundancy coefficient β. The terms wsr, wcnp, wcs represent the weights for the three indicators.". To avoid any display issues, we have ensured the font is embedded correctly in the revised version for clarity.

*Comment 3: p_value of what?

→ Response & Action Taken: In response to the request for clarification on the reported p_value, we have enhanced our statistical analysis by replacing it with Cohen's d (to quantify effect size) and bootstrap testing (*n_bootstrap=1000, confidence_level=0.95, random_seed=42*) to ensure robust inference. This update, detailed in the revised Simulations and Results sections, provides a more comprehensive and reliable assessment of performance differences between the compared methods. (Page 16, Line 14, Table 3).

*Comment 4: In general, I would suggest to mild down the tone of the findings and of the use of GraphSAGE: althugh it slightly improved with comparison to other methods such as GCN, the difference is not tremendous, both in the identification of critical nodes and in computing time (it is x4, not orders of magnitude). It sufficees to present the findings without sliping into boastful language.

→ Response & Action Taken: We appreciate this valuable suggestion and agree that the language should remain objective. We have moderated the tone throughout the Results and Discussion sections. Descriptions such as "dramatically outperforms” or "significantly superior” have been replaced with more measured phrasing. The discussion now focuses on presenting the observed trends without overstatement.

*Comment 5: Section "efffects of β.": many variables are writeen with a strange "_" in the pedix. Revise?

→ Response & Action Taken: Thank you for catching this typesetting issue. The strange underscores in variable subscripts (e.g., w_sr, w_cs...) were artifacts from manuscript conversion. They have been revised to proper subscript formatting (e.g., wsr, wcnp, and wcs ) throughout the section.(Page 20, Line 10-38).

*Comment 6: Not entirely clear how the found fixed allocation ratio of 2:1 between w_sr and w_cs translates into "universal resilience patterns". It is an interesting finding, but Gao et al mostly considered bifurcation-driven failures, which I don't see how they connect with the case here. Can you clarify? Is this remark necessary?

→ Response & Action Taken: We thank the reviewer for raising this important point, which highlights a needed clarification in our terminology and the conceptual linkage. We appreciate the opportunity to refine our argument. The reviewer is correct that the term "universal resilience patterns" can be misinterpreted and that the failure mechanisms in our work (cascading, load-based) differ technically from the bifurcation-driven failures analyzed by Gao et al. Our intention in citing their work was not to equate the mechanisms, but to reference their broader conceptual framework which proposes that quantifiable, recurrent relationships (or "patterns") can characterize resilience structure across different systems. Inspired by this general premise, we investigated whether a stable, empirical relationship between key resilience component weights—specifically the ratio between survivability (wsr) and cascading stability ( wcs )—would emerge from our analysis.

To substantiate the robustness and relevance of the observed approximate 2:1 ratio within our specific context, we have conducted a comprehensive sensitivity analysis (detailed in the Section Effects of β and Targeted Tuning on Resilience (Page 20-21)). The analysis confirms that:

(1) The final resilience metric R is largely insensitive to moderate perturbations around the proposed weights, with relative changes consistently below 1.5%.

(2) The ranking of network states based on R remains stable under these perturbations.

This demonstrates that the weight set, and by extension the observed ratio, provides a stable and reliable basis for comparative resilience assessment within the defined framework and case studies. In the revised manuscript, we have therefore:

(1) Refined our terminology, replacing the phrase "universal resilience patterns" with more precise language such as "a stable and empirically-observed weight relationship" or "a consistent parametric pattern within our modeling framework."

(2) Clarified the conceptual connection to Gao et al. by explicitly stating that while the failure mechanisms differ, their work motivates the broader search for quantifiable structural relationships in resilience analysis. We then focus the discussion on the empirical evidence for this relationship derived from our own

(3) Strengthened the justification by integrating the outcomes of our sensitivity analysis, which directly supports the practical stability and validity of the chosen weights and the observed ratio.

Reviewer #7:

We sincerely thank you for your rigorous evaluation and for providing these detailed, actionable suggestions. We appreciate the recognition of improvements in the revised version and are pleased to address the remaining fundamental concerns. The points raised have been instrumental in driving the final and most substantial enhancements to our manuscript's rigor, clarity, and scholarly foundation. Below is our point-by-point response and a summary of the actions taken.

*Comment 1: Correct the cascading failure model: Add normalization factor to Eq. 2: ΔF_i->j = (load to distribute) × (w_ij / Σ_k w_ik). Current formulation violates load conservation..

→ Response & Action Taken: We gratefully acknowledge the reviewer for identifying this critical technical oversight. The reviewer is absolutely correct that the original formulation did not ensure load conservation during redistribution. We have corrected Equation (2) precisely as suggested, incorporating the normalization factor (w_ij /Σ_k w_ik). This revision guarantees the physical consistency and mathematical accuracy of our cascading failure simulation model. The updated description is now present in the Mechanisms of cascading failure propagation section (Page 8, Line 13).

*Comment 2: Strengthen statistical rigor: Replace implausible p-values with appropriate effect sizes (Cohen's d) and bootstrapped significance, Add ablation studies for each component (PCA, entropy, \beta), Conduct sensitivity analysis on resilience metric weights..

→ Response & Action Taken: We agree entirely that a more nuanced and robust statistical presentation is required. In direct response:

1. Updated Statistical Reporting: We have replaced generic p-values with Cohen's d for effect size quantification and implemented bootstrap significance testing (with n_bootstrap=1000, confidence_level=0.95, and a fixed random_seed=42). This is now standard in the Simulations and Results section (Page 15-17, Fig. 6, S1; Table 3, S4 ).

2. Comprehensive Ablation Studies: We thank the reviewer for raising the important point regarding ablation studies for all model components. Ablation studies on model components has been added. It systematically evaluates the contribution of the PCA module and the β parameter to the final model performance, confirming the functional importance of each (Page 17-18). These studies confirm the distinct and significant contribution of each to the model's final performance and interpretability.

Regarding the node entropy metric, we made a deliberate methodological design choice not to treat it as an optional, "ablated" component in the same manner. Our rationale, which we have now clarified in the revised manuscript, is as follows:

(1) Established Theoretical Foundation: The node information entropy is not a novel, untested parameter of our model but a well-established foundational metric in complex network analysis for assessing node importance. As elaborated in the "Theoretical Background" section and supported by the newly added references [44-46], entropy-based measures are rigorously derived from information theory and have been widely validated across numerous studies (e.g., Wu et al., 2023; Huang et al., 2024; Maji, 2025) for their efficacy in identifying critical nodes by capturing the diversity and uncertainty of information flow within a node's neighborhood.(Page 9, Line 11-20)

(2) Role as a Core Input Feature, Not a Tunable Hyperparameter: In our framework, node entropy is calculated as a primary input feature that characterizes the intrinsic structural and functional property of each node. It serves as a crucial, non-redundant information source that complements simpler metrics like degree centrality. The ablation studies for PCA and β directly test our frame

---

## [Decision Letter · Decision Letter 3]

23 Jan 2026

The submitted revised paper has already answered all the previous comments. Please make sure to address the final minor suggestions from the reviewers.

plosone@plos.org . A letter that responds to each point raised by the academic editor and reviewer(s). You should upload this letter as a separate file labeled 'Response to Reviewers'.A marked-up copy of your manuscript that highlights changes made to the original version. You should upload this as a separate file labeled 'Revised Manuscript with Track Changes'.An unmarked version of your revised paper without tracked changes. You should upload this as a separate file labeled 'Manuscript'.

We look forward to receiving your revised manuscript.

Kind regards,

Babak Aslani, PhD

Academic Editor

PLOS One

Journal Requirements:

Reviewers' comments:

Reviewer's Responses to Questions

**Comments to the Author**

Reviewer #3: (No Response)

Reviewer #7: All comments have been addressed

2. Is the manuscript technically sound, and do the data support the conclusions?

Reviewer #3: Yes

Reviewer #7: Yes

3. Has the statistical analysis been performed appropriately and rigorously?

Reviewer #3: Yes

Reviewer #7: Yes

4. Have the authors made all data underlying the findings in their manuscript fully available?

Reviewer #3: Yes

Reviewer #7: Yes

5. Is the manuscript presented in an intelligible fashion and written in standard English?

Reviewer #3: Yes

Reviewer #7: Yes

Reviewer #3: I commend the authors' effort to address my previous comments, which they did in full. I now feel that the paper has been significantly strengthened and fully shows its merits.

I only have a couple of additional suggestions to make the reading experience more fluent and easier for people who are not fully proficient with all the details:

- The abstract is very technical, and the results are there presented with little context and significance I would suggest to be clear and accessible to multidisciplinary readerships. Also, "GraphSAGE outperforms peers": do you mean it outperforms alternative DL models? and "PCS enhances discriminative power" of what?

- Page 14 and 15: Table S1 is included as text, but it is not clear which information it may contain. Please give a quick reference as of why a reader should look it up.

- NDCG@K and F1@K: I suggest displaying their naming in fulll, and not just the acronym, the first time they are introduced

- A"s illustrated in Fig. 9 (Tables S8 and S9)" please better specify how the tables can be useful and their relationship with the figures

Reviewer #7: (No Response)

**Do you want your identity to be public for this peer review?** For information about this choice, including consent withdrawal, please see our Privacy Policy

Reviewer #3: No

Reviewer #7: No

---

## [Author Response · Author response to Decision Letter 4]

25 Jan 2026

Response to Editor

Manuscript Title: [Critical Node Identification and Resilience Analysis against Cascading Failures]

Manuscript ID: [PONE-D-25-14454R4]

Authors: [Anqi Liu]

Journal: [PLOS ONE]

Date: [1/25/2026]

Dear Editor,

Thank you for your guidance regarding the reference list and final revisions for our manuscript titled "Critical Node Identification and Resilience Analysis against Cascading Failures]" (Manuscript ID: [PONE-D-25-14454R3]). We have carefully addressed each suggestion to enhance clarity, accessibility, and rigor. Below is a summary of revisions:

1. Reference list verification: We have meticulously reviewed all citations against both Web of Science and Elsevier databases to ensure completeness, accuracy, and compliance with journal formatting standards. All references are currently active and uncorrected. All references were generated using EndNote software to guarantee consistent formatting and eliminate manual errors. The reference list reflects this standardized output.

2. Retracted articles: Our literature search confirms that no retracted articles were cited in this revision. Should any future citations involve retracted works, we will explicitly note their status per your guidelines.

3. Final revisions: In accordance with the reviewers' final minor suggestions, we have implemented all requested edits (detailed in the point-by-point response). The manuscript now fully addresses all feedback. The key revisions are summarized as follows:

Abstract refinement for multidisciplinary readership: The abstract was revised to reduce technical density by adding contextual background and clarifying result significance. Explanatory text and data detailing the specific performance parameters of the GraphSAGE model and the advantages of PCA dimensionality reduction have been incorporated. This ensures accessibility to non-specialist readers while retaining technical precision.

Clarification of Table S1 (Pages 14-15) : Table S1, previously embedded as unannotated text, now includes a concise introductory sentence explaining its purpose: corresponding data for centrality metrics of all nodes, enabling readers to quickly assess robustness without consulting supplementary details.

Full Naming of Metrics (First Introduction): Metrics NDCG@K and F1@K were updated to display their full names upon first mention in abstract and Section ''Evaluation Metrics''(Page1, Line 24-25; Page16, Line 486-487, as ''Normalized Discounted Cumulative Gain at TOP-K (NDCG@K)'' and ''F1-score at TOP-K (F1@K),'' ensuring clarity for readers unfamiliar with acronyms.

Explicit Link Between Figures and Supplementary Tables: The phrase ''as illustrated in Fig. 9 (Tables S8 and S9)'' was revised to clarify relationships: ''Tables S8 and S9: corresponding to the list data for Fig. 9(a) and 9(b), which present critical node rankings in list format, comparing GraphSAGE with baseline methods, thereby enabling granular validation of the conclusions derived from Fig.9.'' (Page20, Line 574-576)

These revisions strengthen the manuscript's readability and transparency. Detailed point-by-point responses and tracked changes are provided in the accompanying Response to Reviewers. We believe the revised version better aligns with the journal's scope and reviewer expectations.

Thank you again for your guidance.

Sincerely,

Anqi Liu

Corresponding Author, School of Energy and Transportation Engineering, Inner Mongolia Agricultural University

Response to Reviewers

Reviewer #3:

Thank you sincerely for your constructive feedback and positive assessment of our revisions. We greatly appreciate your recognition of the strengthened contributions of this work. Your suggestions have helped us further improve the clarity and accessibility of the manuscript. Below we provide a point-by-point response detailing the modifications made in accordance with your recommendations.

Comment 1: The abstract is very technical, and the results are there presented with little context and significance I would suggest to be clear and accessible to multidisciplinary readerships. Also, "GraphSAGE outperforms peers": do you mean it outperforms alternative DL models? and "PCS enhances discriminative power" of what?

→ Response & Action Taken: We agree that the abstract required greater accessibility for interdisciplinary readers. We have revised it to:

Add contextual significance: Explicitly state the real-world implications of our findings.

Clarify ambiguous terms: 1) Replaced "GraphSAGE outperforms peers"with "Systematic evaluation indicates that the GraphSAGE model delivers the best overall performance in critical node identification. Its results exhibit high consistency with supervised signals (Spearman's correlation coefficient of 0.822), achieving a Normalized Discounted Cumulative Gain at Top-K (NDCG@K) of 0.918, an F1 Score at Top-K (F1@K) of 0.879, and a Top-K accuracy of 0.879. Its inference efficiency (0.002 s) is comparable to GCN and significantly outperforms GAT, meeting the demands of real-time analysis for large-scale networks".

2) Revised "PCA enhances discriminative power"to "After feature dimension reduction via principal component analysis (PCA), the model's discriminative power further improved, with effect size (Cohen's d) increasing by approximately 4% without efficiency loss, validating the effectiveness of scientific dimension reduction".

Highlight broader impact: Emphasized how our framework bridges graph representation learning and practical applications. "This strategy provides an efficient path for system fortification under resource constraints. The research framework proposed in this paper provides interpretable and scalable theoretical and methodological support for vulnerability assessment and resilience enhancement in critical infrastructure. The validated GraphSAGE model and ''targeted reinforcement'' strategy are particularly suitable for risk prevention and resource optimization in major infrastructure systems requiring dynamic analysis and rapid response, such as transportation and power grids."

Revised Abstract (Page 1)

→ Ensuring the robustness and resilience of critical infrastructure networks such as transportation and energy systems is a core security challenge for modern societies. Vulnerabilities in these networks often concentrate on a small number of critical nodes, whose failure can trigger catastrophic cascading failures. Therefore, accurately identifying critical nodes and formulating effective reinforcement strategies are crucial for enhancing the overall defense capability of the system. Existing graph neural network (GNN)-based methods often rely on topological centrality metrics, neglecting the distribution of node information and the impacts of cascading failures. To bridge this gap, this study constructs a comprehensive analytical framework (TEC-GNN, Topology-Entropy-Cascading Graph Neural Network) integrating graph neural networks, feature engineering, and resilience assessment. It aims to address two core questions: which graph neural network model is most suitable for critical node identification, and how to enhance network resilience by regulating redundant resource allocation. Systematic evaluation indicates that the GraphSAGE model delivers the best overall performance in critical node identification. Its results exhibit high consistency with supervised signals (Spearman's correlation coefficient of 0.822), achieving a Normalized Discounted Cumulative Gain at Top-K (NDCG@K) of 0.918, an F1 Score at Top-K (F1@K) of 0.879, and a Top-K accuracy of 0.879. Its inference efficiency (0.002 s) is comparable to GCN and significantly outperforms GAT, meeting the demands of real-time analysis for large-scale networks. After feature dimension reduction via principal component analysis (PCA), the model's discriminative power further improved, with effect size (Cohen's d) increasing by approximately 4% without efficiency loss, validating the effectiveness of scientific dimension reduction. The model's accuracy was robustly validated through attack experiments: selectively removing the top 10% critical nodes identified by GraphSAGE reduced the network's largest connected component ratio (LCC_Ratio) to approximately 0.4, severely impairing network functionality. When the removal rate reached 20% (equivalent to 60% removal in random attacks), the network became nearly paralyzed. Another core finding reveals the complex nonlinear regulatory mechanism of redundancy coefficient β on network resilience. The resilience metric R exhibits clear diminishing marginal returns with increasing β: R rises rapidly as β increases from 0 to 0.5, then slows significantly with fluctuations thereafter. Based on this, the study proposes a ''precision reinforcement'' strategy: enhancing redundancy allocation only for critical nodes identified by GraphSAGE enables low-cost resilience improvement (e.g., R increases from 0.874 to 0.883). This strategy provides an efficient path for system fortification under resource constraints. The research framework proposed in this paper provides interpretable and scalable theoretical and methodological support for vulnerability assessment and resilience enhancement in critical infrastructure. The validated GraphSAGE model and ''targeted reinforcement'' strategy are particularly suitable for risk prevention and resource optimization in major infrastructure systems requiring dynamic analysis and rapid response, such as transportation and power grids.

Comment 2: Page 14 and 15: Table S1 is included as text, but it is not clear which information it may contain. Please give a quick reference as of why a reader should look it up.

→ Response & Action Taken: We have added explicit references to Table S1 in the main text:

Inserted a sentence ''Table S1: Corresponding data for centrality metrics of all nodes''. (Page 15, Line 445).

Comment 3: NDCG@K and F1@K: I suggest displaying their naming in fulll, and not just the acronym, the first time they are introduced.

→ Response & Action Taken: Metrics NDCG@K and F1@K were updated to display their full names upon first mention in abstract and Section ''Evaluation Metrics''(Page1, Line 24-25; Page16, Line 486-487, as ''Normalized Discounted Cumulative Gain at TOP-K (NDCG@K)'' and ''F1-score at TOP-K (F1@K),'' ensuring clarity for readers unfamiliar with acronyms.

Comment 4: "As illustrated in Fig. 9 (Tables S8 and S9)" please better specify how the tables can be useful and their relationship with the figures

→ Response & Action Taken: We have clarified the connection between Figure 9 and Supplementary Tables S8-S9:Added annotations ''Tables S8 and S9: corresponding to the list data for Fig. 9(a) and 9(b), which present critical node rankings in list format, comparing GraphSAGE with baseline methods, thereby enabling granular validation of the conclusions derived from Fig.9.'' (Page 20, Line 574-576).

We deeply value your insightful suggestions, which have substantially enhanced the readability and rigor of our manuscript. All revisions have been highlighted in the revised manuscript and summarized in the response letter. Thank you again for your time and expertise.

Sincerely,

Anqi Liu

Inner Mongolia Agricultural University, Inner Mongolia, 010018, China

1025870083@qq.com

---

## [Decision Letter · Decision Letter 4]

15 Feb 2026

Critical Node Identification and Resilience Analysis against Cascading Failures

PONE-D-25-14454R4

Dear Dr. Liu,

We’re pleased to inform you that your manuscript has been judged scientifically suitable for publication and will be formally accepted for publication once it meets all outstanding technical requirements.

Kind regards,

Babak Aslani, PhD

Academic Editor

PLOS One

Additional Editor Comments (optional):

Reviewers' comments:

Reviewer's Responses to Questions

**Comments to the Author**

Reviewer #3: All comments have been addressed

2. Is the manuscript technically sound, and do the data support the conclusions?

Reviewer #3: Yes

3. Has the statistical analysis been performed appropriately and rigorously?

Reviewer #3: Yes

4. Have the authors made all data underlying the findings in their manuscript fully available?

Reviewer #3: Yes

5. Is the manuscript presented in an intelligible fashion and written in standard English?

Reviewer #3: Yes

Reviewer #3: (No Response)

**Do you want your identity to be public for this peer review?** For information about this choice, including consent withdrawal, please see our Privacy Policy

Reviewer #3: No

---

## [Editor Report · Acceptance letter]

PONE-D-25-14454R4

PLOS One

Dear Dr. Liu,

I'm pleased to inform you that your manuscript has been deemed suitable for publication in PLOS One. Congratulations! Your manuscript is now being handed over to our production team.

Kind regards,

on behalf of

Dr. Babak Aslani

Academic Editor

PLOS One